# Direct observation of long-range chirality transfer in a self-assembled supramolecular monolayer at interface in situ

Yuening Zhang [1,2], Xujin Qin [1,2], Xuefeng Zhu [3], Minghua Liu [1,3], Yuan Guo [1,2] & Zhen Zhang [1,2] ✉

Due to the interest in the origin of life and the need to synthesize new functional materials, the study of the origin of chirality has been given significant attention. The mechanism of chirality transfer at molecular and supramolecular levels remains underexplored. Herein, we study the mechanism of chirality transfer of N, N′-bis (octadecyl)-L-/D-(anthracene-9-carboxamide)-glutamic diamide (L-/D-GAn) supramolecular chiral self-assembled at the air/water interface by chiral sum-frequency generation vibrational spectroscopy (chiral SFG) and molecular dynamics (MD) simulations. We observe long-range chirality transfer in the systems. The chirality of $C_\alpha$-H is transferred first to amide groups and then transferred to the anthracene unit, through intermolecular hydrogen bonds and π-π stacking to produce an antiparallel β-sheet-like structure, and finally it is transferred to the end of hydrophobic alkyl chains at the interface. These results are relevant for understanding the chirality origin in supramolecular systems and the rational design of supramolecular chiral materials.

Chirality at various hierarchical levels widely exists in nature, and it has received intense attention over the years. Given the significance of symmetry breaking at interfaces, chiral supramolecular self-assembly at interfaces has been one of the most important research fields in recent years; which are closely related to chiral living systems and is also proving to be an essential tool for constructing large functional chiral complexes[1-4]. The mechanism of the self-assembly supramolecular chirality has always been a fascinating scientific topic in this field. It has been found that in living cells, the intrinsic chirality of cells induces the directional rotation of tissue during the formation of tissues, which is a potential mechanism for the morphogenesis of left-right asymmetric tissue[5]. However, how intrinsic chirality induces asymmetric stacking during assembly and how far the chirality transfers from the intrinsic carbon to the self-assemblies remains unclear. In general, the supramolecular chirality of a system is determined by the

chirality of chiral molecules, which is called chirality transfer. Chirality transfer is frequently driven by inter-/intra-molecular interactions such as hydrogen bonding, π–π stacking, van der Waals interactions, and electrostatic interactions[6,7]. Therefore, an analysis of these interactions to understand the mechanism of chirality transfer at interfaces has become an urgent scientific research need[8]. In addition, chirality transfer also has far-reaching implications in many fields, including nanotechnology[9], life sciences[10,11], and the development of new drugs[12] and materials[13]. However, quantitative characterization of the mechanism and distance of chirality transfer still faces great challenges.

Many efficient approaches, such as circular dichroism (CD), vibrational CD, optical rotatory dispersion, and Raman optical activity, can provide chirality-specific spectroscopic information[14]. However, the sensitivity of these approaches is frequently insufficient to identify

[1]Beijing National Laboratory for Molecular Sciences, CAS Research/Education Center for Excellence in Molecular Sciences, Institute of Chemistry, Chinese Academy of Sciences, Beijing 100190, China. [2]University of Chinese Academy Sciences, Beijing 100049, China. [3]Beijing National Laboratory of Molecular Sciences, CAS Key Laboratory of Colloid, Interface and Thermodynamics, Institute of Chemistry, Chinese Academy of Sciences, Beijing 100190, China. ✉e-mail: zhangz@iccas.ac.cn

the chirality of monolayers and thin films. Chiral SFG has proven to be an effective tool for real-time and in situ studies of interfacial chirality in recent years[15–17]. Yan group[18–22] pioneered the use of chiral SFG to investigate complex biomacromolecular structures at interfaces. They identified protein secondary structures (hIAPP)[21] and probed the kinetics of LK$_7\beta$ peptide self-assembly into chiral $\beta$-sheet structures at an interface[22]. Tan et al.[23,24] investigated the structural evolution of hIAPP in negative lipid bilayers. They found that hIAPP evolves through $\beta$-sheet conformers without involving $\alpha$-helical intermediates. Using chiral SFG of the N-H and C$_\alpha$-H groups in the $_L$-LK$_7\beta$ and $_D$-LK$_7\beta$ peptides, Hu et al.[25] found that the chirality of the folded peptides is governed by the C$_\alpha$-H chirality. All these significant studies allowed researchers to better understand the structures, chirality, functions, and interactions of peptides and proteins at biological interfaces. However, understanding the mechanism of chirality transfer of the hierarchical self-assembly of supramolecular structures at the molecular level is still very limited.

Glutamic acid has been widely used as a building block for the fabrication of multiscale chiral structures. It has two carboxylic groups and one amino group, which can form three amide groups and then form inter/intra-molecular hydrogen bonds. Hydrogen bonds between amide groups play a key role in the chiral self-assembly process[26–28].

Here, we use achiral and chiral SFG combine with molecular dynamics (MD) simulation to study the mechanism of chirality transfer in the N, N'-bis(octadecyl)-$_L$-/$_D$-(anthracene-9-carboxamide)-glutamic diamide ($_L$-/$_D$-GAn) supramolecular chiral self-assembled at the air/water interface. We find that in supramolecular assemblies, chiral information from chiral centers can be transferred to hundreds of molecules within 400–500 nm distance between the molecules through weak non-covalent interactions. As far as we know, understanding chirality transfer mechanisms at such a long range in synthetic supramolecular systems remains rare.

## Results and discussion

The $_L$-/$_D$-GAn molecule consists of two hydrophobic alkyl chains, an anthracene unit, and three amide groups (Fig. 1a (Left)), which can easily form intermolecular hydrogen bonds[29]. The anthracene group can easily stack with strong π–π interactions to form the self-assembled nanostructures[30]. Here, we used the chiral SFG (Fig. 1e, g) combined with the compression isotherms (π-A), Brewster angle

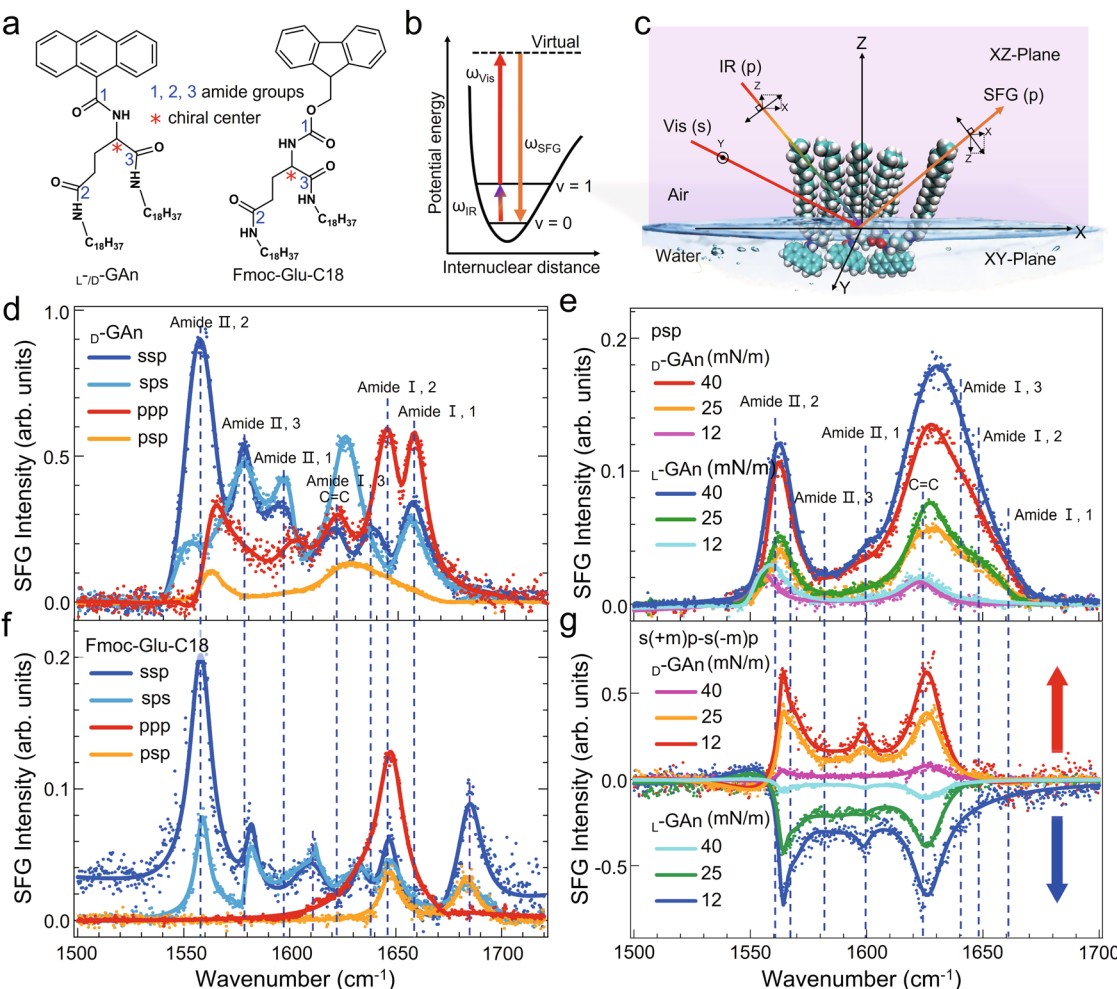

**Fig. 1 | SFG spectral characteristics of assemblies in the amide region. a** The molecular structures of $_L$-/$_D$-GAn and Fmoc-Glu-C18. The sequence numbers (1, 2, 3) represent three amide bonds in different positions of the molecule, and * indicates the asymmetric carbon atom. **b** SFG energy level diagram. Energy level code: Virtual (virtual electronic state), $v = 1$ (vibrational state), $v = 0$ (ground state). **c** A schematic representation of the psp chiral polarization setting to detect the chiral supramolecular monolayer formed by the self-assembly of $_L$-/$_D$-GAn molecules at the interface. (The projection of the electric field of p-polarized and s-polarized light onto the laboratory coordinates (XYZ).) SFG-VS spectra were recorded using achiral ssp, sps, ppp polarization, and chiral psp polarization for **d** $_D$-GAn and **f** Fmoc-Glu-C18 monolayers at a surface pressure of 40 mN/m at the air/water interface. The positions of the three amide peaks in the spectra of the $_L$-/$_D$-GAn molecule (dashed lines) can be clearly assigned by analyzing the amide regions of the spectra of three molecules with similar structures. The chiral SFG-VS was recorded using **e** psp and **g** s(+m)p-s(-m)p polarization for the $_L$-GAn and $_D$-GAn monolayers at different surface pressures at the interface. The assemblies of the $_L$-GAn and $_D$-GAn molecules formed opposite supramolecular chirality at the interface. The fitting results are listed in Supplementary Tables 1–4.

**Table 1 | Peak and β-sheet-like or α-helix-like assignment of amide SFG spectral of $_D$-GAn and Fmoc-Glu-C18**

| Vibration band | $_D$-GAn | | Fmoc-Glu-C18 | |
|---|---|---|---|---|
| Amide I, 1 | 1656.7 ± 0.1 | Disordered structures or β-turn-like | 1684.8 ± 1.3 | α-helix-like |
| Amide I, 2 | 1646.9 ± 0.5 | Disordered structures or β-turn-like | 1647.1 ± 0.5 | Disordered structures or β-turn-like |
| Amide I, 3 | 1637.5 ± 3.1 | Antiparallel β-sheet-like | 1634.3 ± 0.6 | Disordered structures or β-turn-like |
| C=C (Aromatic ring) | 1621.4 ± 1.1 | – | 1609.9 ± 1.4 | – |
| Amide II, 1 | 1598.3 ± 0.2 | – | – | – |
| Amide II, 3 | 1577.6 ± 2.1 | – | 1579.8 ± 1.7 | – |
| Amide II, 2 | 1554.2 ± 2.4 | – | 1555.0 ± 2.3 | – |

microscopy, and MD simulations to provide unique insights into the detailed mechanism of chirality transfer in the $_{L^-/D}$-GAn supramolecular chiral self-assembled at the isotropic air/water interface (the silent sss polarization proves that the interface is isotropic at the molecular level, see Supplementary Fig. 1 for details). The π-A isotherm and Brewster angle microscopy images of $_{L^-/D}$-GAn are shown in Supplementary Figs. 2 and 3.

### Chirality transfer from the chiral center to the amide groups and the anthracene ring

The chiral and achiral SFG spectra (Fig. 1d–g) exhibit multiple peaks attributed to the amide I and amide II modes in the 1500–1700 cm$^{-1}$ region[31]. To verify the details of chirality transfer, the three amide groups connected to the chiral carbon atom must be distinguished and identified for $_{L^-/D}$-GAn monolayer.

Here, Fmoc-Glu-C18 and Fmoc-Asp-C17, which have similar structures (Fig. 1a (Right) and Supplementary Fig. 4a), were used to assign the peaks of the amide groups of $_{L^-/D}$-GAn. We found from Fig. 1d, f, Supplementary Figs. 4 and 5, and Table 1 that most of the peaks are overlapped, and some of the peaks have shifted. The shifted peak positions were assigned to the amide I and II bands of the amide groups due to the different chemical environments. Through two control experiments, we can distinguish the amide I and II bands of the three amide groups of the glutamic acid group. The results are shown in Fig. 1d and Table 1, and the details of spectral peak assignments are written in Supplementary Section 4. Besides, all of the spectra of these three molecules have a peak centered at 1621 cm$^{-1}$, which is attributed to the C=C stretching vibration of the anthracene ring[32].

To further identify the chiral structures of the supramolecular assembly, the achiral (ssp, ppp, and sps) and chiral (s(+m)p-s(-m)p, spp, p(+m)p-p(-m)p, and psp) SFG spectra of the $_D$-GAn monolayer were measured, as shown in Fig. 1d, e, g, and Supplementary Fig. 6. It is proved that the peaks of the amide I band in the achiral and chiral SFG spectra correspond to the secondary structures of peptides and proteins in the literature[33,34]. Since $_{L^-/D}$-GAn molecules are linked differently from standard natural peptide bonds, for clarity, a schematic diagram of the hydrogen bonding links of the $_{L^-/D}$-GAn supramolecular assembly was drawn based on the results of chiral SFG spectroscopy and MD simulations (Supplementary Fig. 7), which exhibit an antiparallel β-sheet-like structure in the $_{L^-/D}$-GAn assembly. Therefore, the peak centered at ~1647 and ~1657 cm$^{-1}$ can be assigned to disordered structures or β-turn-like for $_D$-GAn. The chiral peaks at 1653 and 1647 cm$^{-1}$ in the spectra of Fmoc-Asp-C17 and Fmoc-Glu-C18 (Fig. 1f and Supplementary Fig. 4b, c) were attributed to disordered structures or β-turn-like, and the peak at 1685 cm$^{-1}$ was assigned to α-helix-like.

On the basis of Simpson's symmetry-based theory, the chiral terms of the supramolecular chiral structures of the α-helical structure, parallel β-sheet, and antiparallel β-sheet should be equal (Supplementary Section 1.4)[35]. However, our results (Fig. 1e, g, and Supplementary Fig. 6c) show that the chiral peak positions and bandwidths in the psp, spp, p(+m)p-p(-m)p, and s(+m)p-s(-m)p spectra aren't the same, which indicated that the structural symmetry of the chiral

structure formed among amide groups might be more complicated than that of the α-helical or β-sheet. It can be explained as follows. $_D$-GAn has three amide groups and can form six potential intermolecular hydrogen bond couples: amide 2–3, amide 1–1, amide 2–2, amide 3–3, amide 2–1, and amide 1–3. Therefore, the broader peak at approximately 1630 cm$^{-1}$ should be attributed to amides 2–3 (1637 cm$^{-1}$) and amides 3–3 (1628 cm$^{-1}$), which is the characteristic peak of antiparallel β-sheet-like. The peak shifting and broadening at approximately 1630 cm$^{-1}$ should be due to intermolecular hydrogen bonds (Supplementary Section 5).

Figure 1e, g shows the $_D$-GAn chiral spectral features of the amide II bands in the combination mode at 1550–1580 cm$^{-1}$ via chiral SFG. The $_D$-GAn molecular structure shows that amide 2 is farther from the chiral center than amide 3, and it may largely twist during self-assembly; therefore, the more intense peak at 1554 cm$^{-1}$ should be assigned to amide 2. And the broadness of the peak may be due to more than one peak attributed to the interference of amide 2–2 and 2–3 groups. This is consistent with the notion that there are multiple chiral spectral bands in the amide I range[36,37]. The amide II band of amide 1 near 1598 cm$^{-1}$ shows chiral features under s(+m)p-s(-m)p chiral polarization, but under psp chiral polarization, it shows no chiral features. This indicated that no well-ordered chiral structure, such as helical structures or β-sheets, was formed between amide 1 groups. The chiral spectral features of amide 1 at 1656 cm$^{-1}$ are primarily attributed to the assembly of the anthracene ring through π–π stacking, causing the NH bond to be slightly twisted and thus resulting in chiral spectral features at 1598 cm$^{-1}$.

Furthermore, aromatic groups also generate chiral signals at 1621 cm$^{-1}$ with mirror image peaks (Fig. 1g), indicating that the aromatic ring undergoes asymmetric packing induced by π–π interactions, which can also be confirmed by mirror-symmetric CD signals of the anthracene of the Langmuir-Schaefer films of $_L$-GAn and $_D$-GAn at $^1B_b$- and $^1L_a$-band[38]. The driving force of such asymmetric packing also originated from the chiral center. The $_{L^-/D}$-GAn intensity of the chiral psp peaks of amide II at a surface pressure of 40 mN/m is approximately thirty times greater than that at 12 mN/m (Fig. 1e). This means that the $_{L^-/D}$-GAn monolayers already formed weak chiral self-assembly structures at a surface pressure of 12 mN/m. And the chirality could be larger due to the more hydrogen bonds between the amide groups of the branched glutamic acid core and the stronger π–π stack of the anthracene ring[39]. In addition, we also found (Supplementary Fig. 8a) that the ssp intensity in the amide region of the $_{L^-/D}$-GAn monolayer increases as the surface pressure increases, indicating the amide groups of $_{L^-/D}$-GAn become more ordered at higher surface pressures, which is always true for other monolayers[40] (in Supplementary Section 6 for details). Additionally, we also determined the tilt angle (θ) and twist angle (ψ) of the β-sheet-like segments of $_{L^-/D}$-GAn molecules at a surface pressure of 40 mN/m (Supplementary Section 7).

### Chirality transfer from the chiral center to alkyl chains

Previous research demonstrated that direct chiral transfer at the reaction site often restricts remote stereo control to distances of less

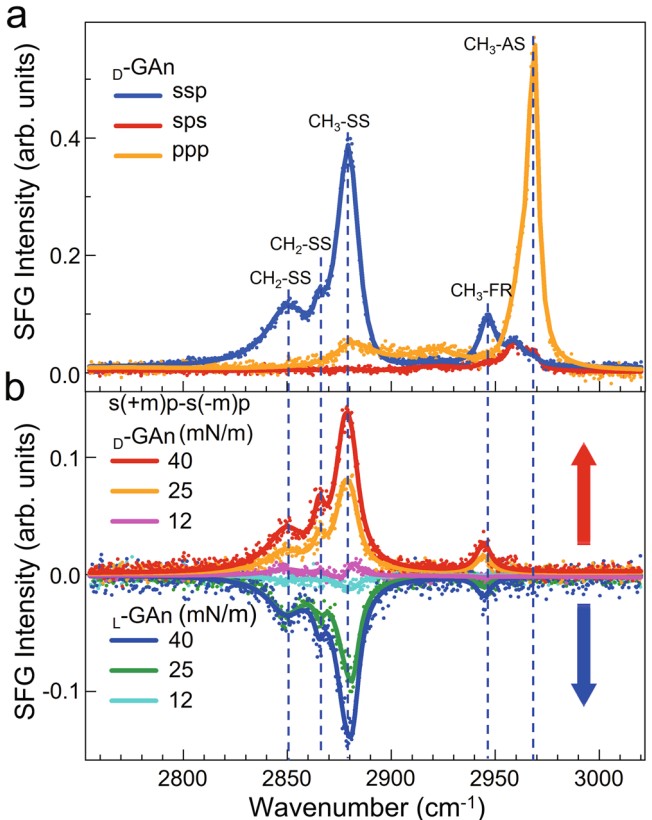

**Fig. 2 | SFG spectra in the region corresponding to C–H stretching vibrations.** **a** Achiral SFG spectra of the D-GAn monolayer at a surface pressure of 40 mN/m at the air/water interface. The fitting results of (**a**) are listed in Supplementary Table 7. **b** Chiral SFG spectra of the L-GAn and D-GAn monolayers at different surface pressures at the interface. The fitting results are listed in Supplementary Table 8. The chirality was transferred to the alkyl chains, and the L-GAn and D-GAn assemblies showed a mirror-symmetrical chiral characteristic peak.

than five bond lengths[41]. We now consider whether chirality can be transferred to more distant groups. Figure 2 shows chiral SFG spectra of the hydrophobic alkyl chains on L-/D-GAn in the C–H region. According to the polarization selection rule, the peaks centered at 2880, 2946, and 2963 cm$^{-1}$ can be assigned to $CH_3$ symmetric stretching ($CH_3$-SS), $CH_3$ Fermi resonance ($CH_3$-FR), and $CH_3$ asymmetric stretching ($CH_3$-AS), respectively; while the peaks centered at 2850 and 2865 cm$^{-1}$ are attributed to the symmetric stretching mode of $CH_2$ groups ($CH_2$-SS)[42]. Based on the principle of the inductive effect[43], the peaks centered at 2850 and 2865 cm$^{-1}$ can be attributed to the symmetric stretching of $CH_2$ groups in the alkyl chain and glutamate unit. However, neither the chiral SFG signal nor the achiral SFG signal of the =C-H stretching vibration (central peak at 3046 cm$^{-1}$) of the anthracene group of the L-GAn and D-GAn molecules could be observed. The reasons are discussed in Supplementary Section 8. In addition, we also observed that the intensity of the $CH_3$-SS peak (2880 cm$^{-1}$) increased as the surface pressure increased, indicating that the conformation of the L-/D-GAn alkyl chains was more ordered and formed a well-organized supramolecular structure (Supplementary Fig. 8b)[44–47]. The morphology of the D-GAn Langmuir monolayers at various surface pressures detected by Brewster angle microscopy (Supplementary Figs. 2 and 3) also confirms the well-organized supramolecular structure.

More importantly, we observed the interference chiral SFG spectra with s(+m)p-s(-m)p polarization of L-/D-GAn at the air/water interface in the C-H region with a mirror image, shown in Fig. 2b. This phenomenon for $CH_3$ groups in biosystems has been observed via

chiral SFG[16]. In a recent report, a chiral signal of $CH_3$ detected using chiral SFG should attribute to the presence of intermolecular coupling[15]. In our current situation, we obtained the interference s(+m) p-s(-m)p spectra, but no pure chiral spectra of psp and spp were obtained (Supplementary Fig. 11). This is because the chiral terms of the $CH_3$ group are generally much smaller than the achiral terms. The peaks at 2850, 2865, 2880, and 2944 cm$^{-1}$ all show mirror images for L-GAn and D-GAn. This suggested that once the different chirality of the chiral centers of L-GAn and D-GAn molecules is transferred via the glutamic acid unit at the interface, the chirality is transferred from the asymmetric carbon atoms to the alkyl chains. In Fig. 2b, more intense chiral SFG signals were observed as the surface pressure increased, indicating that the methyl groups were packed together, which strengthened the intermolecular coupling[15,48]. The global-fitting parameters for the methyl groups in the achiral SFG spectra of the L-GAn and D-GAn monolayers in Supplementary Tables 6 and 7 indicate that the alkyl chains in the L-/D-GAn enantiomers have the same interfacial orientations with respect to the interface normal. By comparing the orientations determined by theoretical simulation, as shown in Supplementary Fig. 12, we found that the orientation angle changed from approximately 44° to 37° for the methyl groups of L-GAn and D-GAn molecules as the surface pressure increased[49–51].

## Exploration of chiral supramolecular structures and intermolecular interactions at interfaces via MD simulations

Here, we performed MD simulations under NPT and periodic boundary conditions with GROMACS. Based on these MD results, we also performed orientation and hydrogen-bond analyses (Experimental Section and Supplementary Section 10).

Figure 3a shows a snapshot of the initial system. Figure 3b, c shows the front view and side view of the equilibrium state of the L-GAn monolayer at the interface. The simulations clearly demonstrated the process of L-GAn molecules from spreading at the interface to forming an ordered monolayer at 53 Å$^2$ (surface pressure of 20 mN/m). As seen in these figures, 100 L-GAn molecules form an ordered monolayer at the interface, with the alkyl chains pointing obliquely to the gas phase, and the anthracene ring and amide groups are exposed to water.

Figure 3d shows the number of intermolecular hydrogen bonds formed between the different amide groups of the L-GAn monolayer. For example, amide 2 of one L-GAn molecule is most likely to form a hydrogen bond with amide 3 of the other L-GAn molecule, and hydrogen bonds can also be formed by amide 1–1 and amide 2–2 between neighboring molecules. However, almost no amide 3–3, 2–1, and 1–3 hydrogen bonds could form between the amide groups of nearby L-GAn molecules. This difference in hydrogen-bonding capacity provides crucial insights into the formation of the antiparallel β-sheet-like structures that create macroscopic chirality. Moreover, the total number of hydrogen bonds formed between L-GAn molecules increased from 108 to 120 (Supplementary Fig. 13d), indicating that the lateral interaction of the L-GAn molecules was reinforced during the self-assembly process.

To visualize the π−π stacking and twisting of the anthracene ring, four anthracene molecules are illustrated in Fig. 3f. The anthracene ring twists in a fixed direction to form stacking structures with the next anthracene ring. Such twisted stacking causes the C=C stretching peaks of the anthracene ring observed in the chiral SFG spectra in Fig. 1e, g. The stacking structure of the anthracene groups can be better understood by analyzing the orientation angle and the orientation contribution of the anthracene ring. The statistical analysis of the orientation angle of the anthracene ring was proposed to be 70°, with an orientation contribution of 0–160°, as shown in Fig. 3f. Such a large orientation distribution also indicated the possibility that the anthracene rings stack in a certain direction with a certain twist angle that exhibited chirality. Figure 3e shows that the orientation angles of the two long alkyl chains mainly fall within approximately 40°, which is

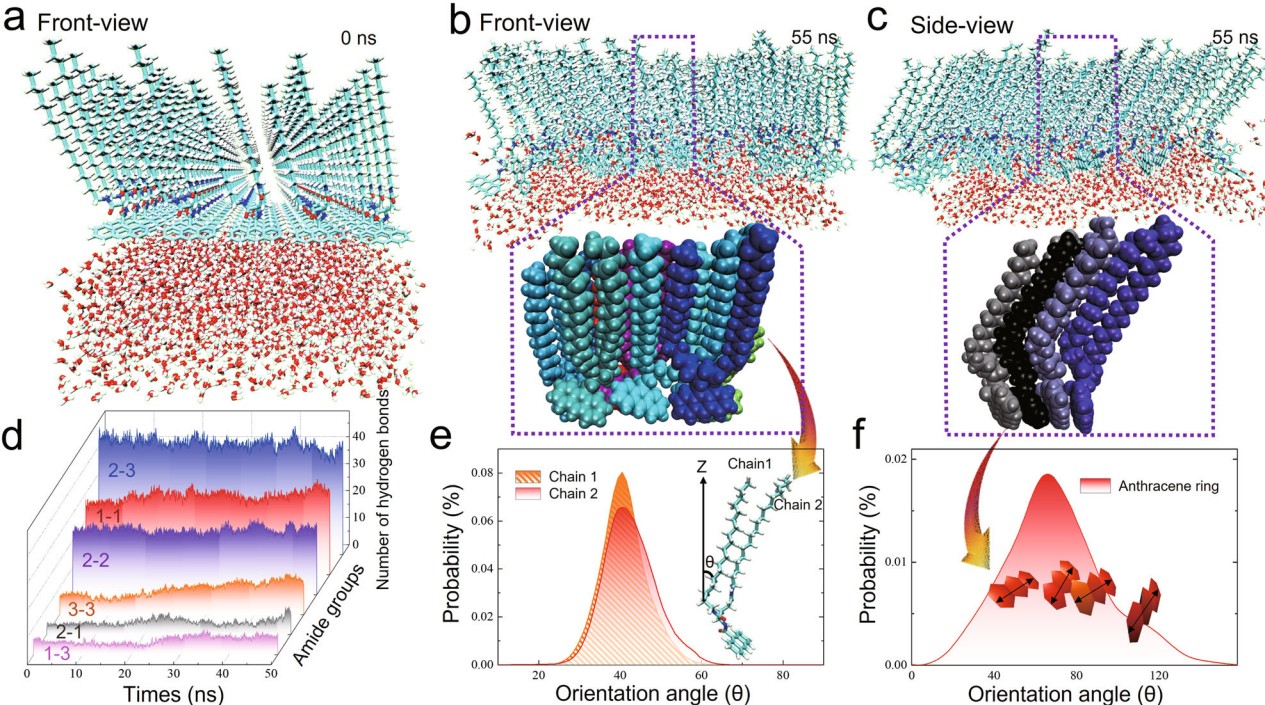

**Fig. 3 | MD simulation results of chiral supramolecular structures formed by the self-assembly of L-GAn molecules at the air/water interface. a** Front-view of the initial system snapshot of the L-GAn simulation system (for clarity, only 50 L-GAn molecules and some water molecules are shown). **b** Front-view, and **c** side-view system snapshots of the last frame after a 50 ns MD simulation of the L-GAn simulation system at the air/water interface (100 L-GAn molecules and some water molecules). To clearly observe the assembly arrangement of the L-GAn molecules, the last frame of several molecules located in the central area of the box is also shown (snapshots of all L-GAn and water molecules are shown in Supplementary Fig. 13b, c). **d** Statistical graph of the change in the number of hydrogen bonds formed by the different amide groups (amide 1–3) between adjacent molecules with the simulation time. Orientation distribution statistics of (**e**) two alkyl chains and (**f**) an anthracene ring. The inset in (**e**) shows only an L-GAn molecule from (**b**). The orientation angle of the alkyl chain is described by the angle ($\theta$) between the average orientation of all carbon atoms of the alkyl chain and the interface normal (black arrow). The inset in (**f**) shows only the anthracene ring from (**c**), and the other L-GAn molecular groups are hidden. The dipole direction of the anthracene ring is shown by the arrow.

consistent with the orientation angles of the alkyl chains. In addition, we confirmed that this conclusion from SFG analysis in the above experiments would not be affected by magic angle (Supplementary Section 9).

**Quantification of the long-range chirality transfer from intrinsic molecular chirality to macro-self-assembly chiral structures**

According to the chiral SFG and MD simulation results, as well as the orientation and number of hydrogen bonds, we can infer the chiral supramolecular assembly details of L-GAn molecules (Fig. 4). L-GAn molecules self-assemble at the air/water interface to form nanorod structures (Supplementary Fig. 14a, b and Fig. 4a) where each nanorod is assembled from multiple antiparallel $\beta$-sheet-like structures (Fig. 4b). The L-GAn molecules stack along the x-axis to produce the width of the nanorods and along the y-axis to form the length of the nanorods due to the steric hindrance of the strong intermolecular hydrogen bond of amide 2–3 and the twisting of the anthracene ring at the interface. Here, we define the nanorods' length, width, and height along the y-axis, x-axis, and z-axis directions, respectively, as shown in Fig. 4c. We then determined the thickness of the L-GAn monolayer at equilibrium from MD simulation to be approximately 36 Å (Supplementary Fig. 13e), which is very close to about 29–43 Å height of nanorod in the z-axis direction from AFM[38] and is also approximately the height of one molecule. In the same way, we determined the width of the nanorod, about ~30 molecules along the x-axis; and the length, which is ~900 molecules along the y-axis, as shown in Fig. 4c in three dimensions. Compared with L-GAn molecules, the Fmoc-Glu-C18 molecule is connected to the glutamic acid unit through a flexible chain to produce steric hindrance, which may weaken the $\pi$–$\pi$ stacking

of aromatic rings and the intermolecular hydrogen bond, so that it forms shorter nanorod structures (Supplementary Fig. 15). The detailed quantitative number of molecules composing the Fmoc-Glu-C18 nanorods has been discussed in Supplementary S11. In this hierarchical assembly process, the chirality is transferred from the chiral center to the amide groups, the anthracene ring, the alkyl chains, the methyl groups at the end of the alkyl chains, and finally the whole macro-self-assembly chiral structure (Fig. 4d).

In conclusion, our work provides insight into the long-range chirality transfer in the L-/D-GAn supramolecular chiral self-assembled at the air/water interface by using SFG and MD simulations. We proposed that molecular chirality is transferred to supramolecular assemblies through the following mechanism: the chiral centers located at the glutamate unit induce slight twists of the amide groups connected to it when via intermolecular hydrogen bonds, transferring chirality to the amide groups, and then the chirality is transferred to the anthracene and hydrophobic alkyl chains. This will eventually result in a complex assembly structure with an antiparallel $\beta$-sheet-like structure at the interface. The chirality direction is determined by the chiral center, which makes up the entire supramolecular chiral structure of L-GAn self-assembly at the interface. These findings have significant implications for understanding the relationship between intrinsic molecular chirality and supramolecular chirality and can help us rationally design and fabricate chiral nanostructures.

## Methods
### Sample preparation
The synthesis and characterization of the molecules (L-/D-GAn) were published elsewhere[52] (Supplementary Section 14). Chloroform and

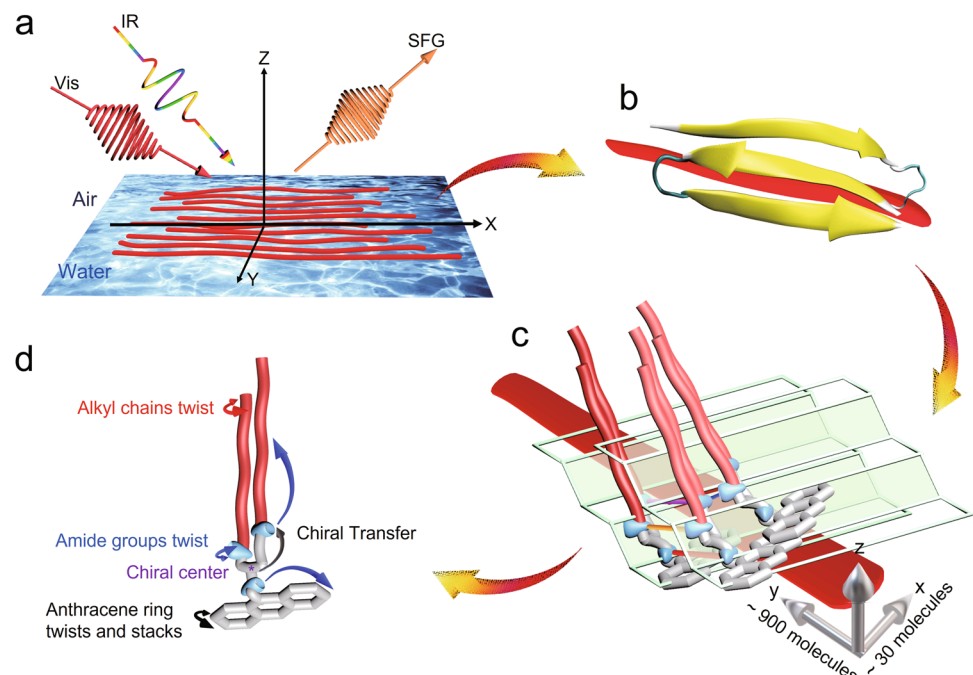

**Fig. 4 | Schematic diagram of the supramolecular assembly arrangement and chirality transfer mechanism. a** A schematic representation of SFG to detect the nanorod structures formed by the self-assembly of $_L$-/$_D$-GAn molecules at the interface. **b** $_L$-/$_D$-GAn molecules assemble at the interface to form antiparallel β-sheet-like structures and further assemble into nanorod structures. **c** Schematic diagram of the intermolecular interactions and hierarchical assembly process of $_L$-/$_D$-GAn molecules. The nanorod contains about 30 molecules along the $x$-axis, about 900 molecules along the $y$-axis, and one molecule along the $z$-axis (alkyl chain, red; amide group, blue; virtual β-sheet-like plane, green). The hydrogen bond is formed by the amide groups on two adjacent $_L$-GAn molecules (amide 2 and amide 3, blue; amide 1 and amide 1, red; amide 2 and amide 2, purple; amide 3 and amide 3, orange). **d** Chirality is transferred from the chiral center to the amide groups. The hydrogen bonds between the amide groups cause the amide groups to twist and further transfer the chirality to the anthracene ring and alkyl chains. The twisting of the alkyl chains allows chirality to be transferred to the terminal methyl group.

---

ultrapure water were used as the spreading solvent and subphase for the $_L$-/$_D$-GAn monolayers, respectively. Chloroform was purchased from Beijing Chemical Works. Ultrapure Milli-Q water (Millipore, 18.2 MΩ cm) was used in all cases.

## Surface pressure-area isotherm measurements

The surface pressure–area (π–A) compression isotherms were measured by a computer-controlled Langmuir trough ($A_{total}$ = 273 cm²; KSV Mini Trough; Biolin Scientific, Espoo, Finland). The trough was made of Teflon and equipped with two compression barriers made of Delrin (DuPont, Wilmington, DE). The surface pressure of the Langmuir monolayer was monitored by a Wilhelmy plate made of platinum and flamed with a Bunsen burner before use. The trough and barriers were rinsed with ethanol and Millipore water (Millipore, Billerica, MA) several times before the measurements. The whole trough was enclosed in a homebuilt Plexiglas protective box to eliminate dust and other possible airborne contaminants. Before spreading the $_L$-/$_D$-GAn solution on pure water, the aqueous surface was swept by the barriers to ensure surface cleanliness, and cleanliness was confirmed by a final surface pressure of <0.2 mN/m. When preparing the Langmuir monolayers, an appropriate volume of the $_L$-/$_D$-GAn chloroform solution ($5 \times 10^{-4}$ M) was spread onto the surface of the aqueous subphase (18 MΩ cm, 25 °C) with a microsyringe in a dropwise manner to form air/water interfacial assemblies. After allowing the solvent to evaporate for 15 min, the surface pressure-molecular area (π-A) isotherms were recorded by compressing the floating film at a rate of 7.5 cm²/min at 25 °C. In addition, the structural change of $_L$-/$_D$-GAn Langmuir under different surface pressure–molecular areas was also monitored in situ by Brewster angle microscopy. Brewster angle microscopy images were trimmed and toned to enhance the contrast.

## Chiral SFG-VS measurements

The SFG-VS setup has been reported previously[53]. In brief, the SFG-VS system contains two sets of electronically synchronized (Tsunami 3950 C, Spectra-Physics) Ti sapphire oscillators/amplifiers with a 35-fs pulse width (Mai-Tai SP and Spitfire Ace, Spectra-Physics) operating at a 1 kHz repetition rate at a fundamental of 800 nm, which provides a broadband spectral infrared (IR) beam through an optical parametric amplifier (OPA, TOPAS Prime, Light Conversion). Subwavenumber spectral resolution was provided by a visible (VIS) beam coming from the other set with an ~50 ps pulse width (Tsunami, Spectra-Physics). The visible pulse profile is a Gaussian profile in the time domain. The visible and IR beams overlapped spatially and temporally at the sample, with incidence angles of 63° and 40°, respectively. Each SFG spectrum was acquired using a monochromator (Acton SP2750, 750 mm, 1200 lines/mm grating, 200 nm slits) and EMCCD (Princeton Instrument, PRO-1600×400BX). For the spectra in the C−H (IR power = 12 mW, Vis power = 120 mW) and amide (IR power = 5 mW, Vis power = 180 mW) regions, the acquisition times were 20 and 30 min, respectively. In the C−H and amide regions, the gain was 1 and 300, respectively. All SFG measurements were carried out at the air/water interface at surface pressures of 12, 25, and 40 mN/m. During the measurement, the sample was rotated at a speed of 10°/s to reduce the laser heating effects. The effective second-order susceptibility ($\chi^{(2)}_{eff}$) can be expressed as linear combinations of the second-order susceptibility tensor elements ($\chi^{(2)}_{IJK}$), and the expression under achiral ssp, sps, and ppp polarization and chiral psp and spp polarization are provided in Supplementary Section 1. We measured the interface chirality in the amide region with psp, spp, s(+m)p-s(-m)p, and p(+m)p-p(-m)p polarization. Moreover, because the chiral signal was too weak to be detected with psp polarization in the C−H region, s(+m)p-s(-m)p polarization was used to quantify chiral SFG-VS in the C−H region. The

interference crossing term for s(+m)p-s(-m)p, is[54]

$$I_{s(+m)p} - I_{s(-m)p} \propto \mathrm{Re}\left\{\chi^{(2)}_{ssp}\chi^{(2)*}_{spp}\right\} \tag{1}$$

## All-atom MD simulations

In this work, the CHARMM36 all-atom force field[55] and GROMACS software[56] (version 2019.4) were used for all simulations. Parameters for $_L$-GAn molecules were obtained using the CGenFF module of the CHARMM-GUI web server[57,58]. A starting structure of 100 $_L$-GAn molecules was placed into a $100 \times 100 \times 200$ Å box, half of which was filled with 6242 TIP4P water molecules and 0.15 M Na$^+$ and Cl$^-$ ions to neutralize the system to create a vacuum-water interface. Initially, we limited the orientation of the anthracene ring. The area per $_L$-GAn molecule quickly decreased from 100 to 53 Å$^2$ within 2 ns of simulation, and the energy of the system was minimized, followed by equilibration in an NVT ensemble for 3 ns at 298 K. A Berendsen thermostat with a 1 fs time step was used to maintain the temperature[59]. To better quantify the statistical properties of $_L$-GAn molecules, the simulation system released the orientation restriction and was further run for 50 ns under the NVT ensemble. Finally, the orientation angles of the alkyl chains and benzene rings in the $_L$-GAn molecules were calculated using our in-house script over this 50 ns MD trajectory. During the calculations of the orientation angles of the alkyl chains, each carbon chain was divided into two parts (each containing 9 C atoms), and the coordinates of the two parts $(x_1, y_1, z_1)$ and $(x_2, y_2, z_2)$ were calculated to obtain a vector that could reflect the orientation of the carbon chain. The angle ($\theta$) between the vector and the normal direction of the interface was calculated. System snapshots and movies were generated by VMD[60].

## Data availability

All data, including SFG spectra; MD simulation; AFM, BAM; surface pressure-isotherm data and analysis of orientation generated in this study have been deposited in the Science Data Bank under accession code https://doi.org/10.57760/sciencedb.06698.

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

## Acknowledgements

Z.Z. and Y.G. are grateful for funding from the Natural Science Foundation of China (NSFC 91856121, 22173112, and 21873104), Z.Z. thanks the Beijing Natural Science Foundation of China (2202067), Chinese Academy of Sciences for support (Grant No. YJKYYQ20180014). We thank Prof. Li Zhang in ICCAS for thoughtful discussion, for the spectral assignment of amide bands.

## Author contributions

The research project was initiated by Z.Z., M.H.L., and Y.G.. X.F.Z. synthesized the molecules and purified them. X.J.Q. calibrated the optical path for the SFG measurement and carried out SFG instrument repair and maintenance. Y.N.Z. conducted the spectroscopy experiments and MD simulations. Y.N.Z and Z.Z analyzed the experimental data. The manuscript was prepared and written by Y.N.Z and Z.Z. and was approved by all coauthors.

## Competing interests

The authors declare no competing interests.
