## [Peer Review File · Nature Communications]

Direct Observation of Long-Range Chirality Transfer in a Self-Assembled Supramolecular Monolayer at Interface in SituEDITORIAL NOTE: Figure B1 in this Peer Review File has been amended to remove third-party material where no permission to publish could be obtained.

REVIEWER COMMENTS

Reviewer #1 (Remarks to the Author):

In this manuscript, the authors investigate the chirality transfer mechanism at molecular level and supramolecular level. Understanding the mechanism of chirality transfer behaviors in supramolecular assemblies not only is of great importance in the aspect of better understanding to the relationship between supramolecular chirality and molecular chirality, but also provides a theoretical basis in designing and constructing chirality assemblies. This work would provide insight into long range chirality transfer in the L-/D-GAn supramolecular assemblies at the air/water interfaces by using of sum-frequency generation vibrational spectroscopy and molecular dynamics simulations. Importantly, this paper has designed suitable experiments and concluded the chirality transfer process from the carbon center in the glutamate unit to the amide groups, and finally to the anthracene and hydrophobic alkyl chains. The theoretical conclusion was consistent with the published experimental results and explained the widely existed chirality transfer and magnified process in supramolecular fields and in nature. Therefore, I would recommend this manuscript to be published on Nature Communications after the following revisions:

- 1) Considering that the author used published results (Figure S11) to clarify the accuracy of the theoretical conclusions, which is of great importance. Considering that the AFM images are experimental results of assembled L-GAn, if possible, please provide AFM images of assembled Fmoc-Glu-C18 to further prove the theoretical conclusions for the other assembly block.
- 2) Getting the theoretical results is further confirmed with experiential data making the conclusion more valid. Is it possible for the authors to investigate the CD spectra of relevant assembled systems to verify the chirality transfer results based on SFG-VS spectra?
- 3) In Page 11, the authors determined the number of molecules on height (~30 molecules. I feel it should be height rather than thickness, which can distinguish from the approximate thickness of the monolayer), width, and length based on approximate thickness of monolayer from the width. However, one molecule usually exhibits different molecular lengths on the three dimensions, so the approximate thickness of monolayer should not be used directly to determine the molecular number on the width and length. Therefore, the author should clearly show the method or explanation for the determination. Single crystal data or at least optimized calculated molecular structures are suggested to be used for the determination.
- 4) There are too many abbreviations in the manuscript, which may confuse readers. Some of them do not appear frequently, which is not necessary to use abbreviations. Please check and reorganize it as appropriate.
- 5) Regarding the organization of the Figures: a) In Figure 3, please adjust the arrows to avoid characters being hidden by dark red arrows. b) Figure 4 is too small for reading, please reorganize and amplify it.
- 6) About the TOC figure: The labels of the spectra are hidden by the graphical representation. It is not readable. Also, please reorganize the TOC scheme for better illustration of the significance of this paper.
- 7) The authors may consider citing the references related to diverse chiral self-assemblies, e.g., Adv. Mater. 2020, 32, 1801335; Chem. Sci. 2020, 11, 9989; ChemPhotoChem, 2022, 6, e202100256.

Reviewer #2 (Remarks to the Author):

The theory for this manuscript may need some further consideration.

1. SPP/PPS/PSP polarization SFG signals contain only chiral signals under the assumption that the molecules or self-assemblies are isotropic, i.e., $\chi_{XXX}=\chi_{YYY}=\chi_{XXY}=\chi_{XYX}=\chi_{YXX}=\chi_{YXY}=\chi_{YYX}=\chi_{XYY}=\chi_{YZZ}=\chi_{ZYZ}=\chi_{ZZY}=0$. Or a non-chiral system could have SPP/PPS/PSP signal as well. For example, some nonlinear crystals do

not have chirality but have SPP/PPS/PSP signals, or please check Prof. Andre Knoesen's Biophysical Journal article in 2007 (table 1). This isotropic assumption is usually true for short chain SAMs samples but may not be true for systems with strong intermolecular interactions, since they can form crystals or microcrystals. For example, Prof. Andre Knoesen reported an example on Biophysical Journal in 2007 and Prof. Wei Xiong reported an example on JPCB in 2019.

Even though the authors rotated the sample, the SFG signals generated at different time are not coherent and will not cancel out. Therefore, the measured effective second-order susceptibilities still contain anisotropic terms.

If the sample is anisotropic, probably what the authors observed in this manuscript was not long-range chirality transfer, but was because these molecules formed a microcrystal as a whole.

To check this, the authors should have measured SSS polarization signal. If SSS polarization signal is non-zero, it means anisotropic interface and their data analysis needs to rewrite. If SSS polarization signal is zero, their analysis could be correct.

2. The tilt angle calculation method used in this manuscript is under 2 assumptions:

a. The same assumption as in 1. If the system is anisotropic, more complicated analysis and calculation will be needed.

b. The assumption that the tilt angle distribution is narrow, i.e., $\langle \cos^3(\theta) \rangle / \langle \cos(\theta) \rangle = \langle \cos^2(\theta) \rangle$. As Prof. Garth Simpson and Prof. Kathy Rowlen pointed out on a JACS article in 1999, this assumption may not be true and could cause big trouble if the tilt angle is calculated to be around 39.2°, which is called magic angle. Therefore, Prof. Simpson and Prof. Rowlen claimed that if the calculated tilt angle is around 39.2°, especially 39±2°, special treatment would be needed. In this manuscript, the tilt angles were calculated to be 44° and 37°, while the authors did not provide special treatment in this manuscript.

3. The author claimed "In addition, we also found (Supplementary Fig. 6a) that the ssp intensity in the amide region of the L-/D-GAn monolayer increases as the surface pressure increases, indicating the amide groups of L-/D-GAn become more ordered at higher surface pressures, which is always true for other monolayers". However, when monolayer surface pressure increases, the surface area (per molecule) should decrease, which indicates a higher molecular density per area. Therefore, I wonder if the authors considered the relationship between SFG signal intensity vs. molecular density.

Reviewer #3 (Remarks to the Author):

The authors study a monolayer of supramolecular self-assemblies of a chiral amphiphilic molecule at the air/water interface. They use methods of chiral sum frequency generation (SFG) vibrational spectroscopy and molecular dynamics (MD) simulation. The chiral amphiphilic molecule has two C18 carbon chains and one aromatic anthracene functional group. These three groups are connected to a small linker backbone via three amide groups. One of the amide groups that links to anthracene is connected to a carbon atom that is a chiral center. The authors took chiral SFG spectra of the enantiomers of this molecule at the air-water interface. The SFG spectra show that the carbon chains of the two enantiomers give the same chiral SFG response for the methyl and methylene stretching modes but with opposite signs. Moreover, the authors also assign the vibrational C=C stretching modes of anthracene and observe these modes are also with opposite phase for the two enantiomers. The authors therefore conclude that chirality is transferred from the chiral-center carbon atom to the two carbon chains and to the anthracene group. This major conclusion is supported by experiments. Because the long-range chirality transfer is of interest to many research fields, the manuscript should be accepted for publication if the authors can fully address the following major questions and concerns:

Major concerns:

1. The major observations to support the main point is that the enantiomers give the chiral SFG

response of the methyl and methylene stretches with opposite phase. However, in the manuscript, the authors first discuss the interpretation and assignments of amide I and amide II modes in order to assign the C=C stretches of anthracene. Although these assignments need to be done, it is not crucial to the main theme of the paper – chirality transfer from the chiral-center C atom via long distance (2-3 nm) to the methyl and methylene groups on the carbon chains. Thus, the authors may want to reduce the emphasis of the assignments and interpretations of the amide spectra.

2. In interpreting the amide spectra, the authors discuss the data in the context of protein secondary structures. This can be another distraction. In this study, although the molecule under study contains three amide functional groups, they are not connected in the same way as standard native peptide bonds in a protein. Hence, the molecule is not likely to form standard protein secondary structures by itself or via self-assembling, especially that the conformation of the molecule should be constrained by the amphiphilic air-water surface. Therefore, it can be problematic to discuss how the molecule can form beta-sheet and alpha-helical structures. Indeed, if one wants to make the points about forming these types of secondary structures, analysis of Ramachandran angles is typically needed. However, as the interpretations and assignments of the amide spectra do not impact the major conclusion (i.e., long-range chirality transfer), the authors may as well simplify the discussion and move some of the interpretation to the supplementary information.

3. It is like that the authors spent a lot of effort to interpret the amide spectra because they need to assign the anthracene C=C stretches. The frequency of the aromatic C=C stretches falls into the amide I region. Thus, to solidly show chirality transfer from the chiral-center carbon to the anthracene group, the authors need to not only assign the C=C stretches but also all the other amide bands. Nonetheless, the authors can also consider probing the aromatic C-H stretches of the anthracene group, which should be at frequency $> 3000 \text{ cm}^{-1}$. Perets et al. (Langmuir, 2022) showed recently that aromatic bases in DNA show chiral C-H stretches at frequency $> 3000 \text{ cm}^{-1}$ upon hybridization. If the anthracene groups also form chiral assemblies, they may also show chiral SFG response of aromatic C-H stretches.

4. The authors propose a structural model of supramolecular assemblies (Figure 4). In the model, the chiral amphiphilic molecules form beta-sheet structures. For the reason discussed in point#2, it is unclear how the beta-sheet structure is formed. If the authors want to make the claim, they should at least build a molecular model that forms beta-sheet structures using MD and then analyze the Ramachandran angles. This additional evidence is necessary because the amphiphilic molecule can have orientational constraints at the interface and these orientational constraints can hinder formation of beta-sheet structures.

5. Although the chirality transfer reported in the current study is impressive, the authors need to provide additional backgrounds in the Introduction about long-range chirality transfer. Is it the first observation of chirality transfer over 2-3 nm? Some background information may help highlight the novelty and significance of the work that can justify publication in Nature Communications.

Other suggestions:

a. Page 4: "inter/Intra hydrogen bonds." The term "intra" should be in lower case.

b. Page 4: "pi-A." The authors may want to define pi and A for readers who are not familiar with surface pressure measurements.

c. Page 9: "Fig.3a" and "Fig. 3d" should be "Figure 3a" and "Figure 3d."

d. Figure 2b: "s+/-mp" should be written out as the actual physical quantity being measured, i.e., difference between (s + mp) and (s - mp). The current label could be confusing especially to readers who are not in the SFG field.

Response to Reviewers

We thank the Reviewers for carefully reading the manuscript and for the valuable suggestions. In the revision of this manuscript, we took these suggestions into full consideration and did additional experiments to answer the questions raised by the reviewers. In addition, we also changed and supplemented some of the discussions in the manuscript as suggested by the reviewers. Due to the space limitation of the revised manuscript, some of the results and discussions of the additional experiment are in the supplementary information. The revision has been highlighted in the response letter, the revised main text, and the supporting information. We hope this revision will satisfy the reviewer's concerns. The major revisions and new analyses we have undertaken are summarized below.

1. Performed the additional AFM measurements of Fmoc-Glu-C18 Langmuir-Blodgett films and estimated the number of Fmoc-Glu-C18 molecules forming nanorods (Fig. A1, shown in Supplementary Fig. 15) in the supplementary.
2. Discussed the CD spectra of L - D -GAN assembled systems to verify the chirality transfer based on SFG-VS spectra (Fig. A2).
3. Performed the additional sss polarization spectra to confirm our hypothesis that the self-assemblies are isotropic at the interface (Fig. A4, shown in Supplementary Fig. 1).
4. Confirmed the orientation angles of the terminal alkyl groups are not affected by the magic angle (Fig. A5, shown in Supplementary Fig. 12).
5. Redefined the Ramachandran angle in the supramolecular system and proposed a secondary structure-like conformational in the supramolecular system. (Fig. A6, shown in Supplementary Fig. 7).

Below, we give a point-by-point response to the comments of the referees.

Reviewer #1:

In this manuscript, the authors investigate the chirality transfer mechanism at molecular level and supramolecular level. Understanding the mechanism of chirality transfer behaviors in supramolecular assemblies not only is of great importance in the aspect of better understanding to the relationship between supramolecular chirality and molecular chirality, but also provides a theoretical basis in designing and constructing chirality assemblies. This work would provide insight into long range chirality transfer in the L - D -GAN supramolecular assemblies at the air/water interfaces by using of sum-

frequency generation vibrational spectroscopy and molecular dynamics simulations. Importantly, this paper has designed suitable experiments and concluded the chirality transfer process from the carbon center in the glutamate unit to the amide groups, and finally to the anthracene and hydrophobic alkyl chains. The theoretical conclusion was consistent with the published experimental results and explained the widely existed chirality transfer and magnified process in supramolecular fields and in nature. Therefore, I would recommend this manuscript to be published on Nature Communications after the following revisions:

1) Considering that the author used published results (Figure S11) to clarify the accuracy of the theoretical conclusions, which is of great importance. Considering that the AFM images are experimental results of assembled L-GAn, if possible, please provide AFM images of assembled Fmoc-Glu-C18 to further prove the theoretical conclusions for the other assembly block.

Author reply: We appreciate the referee's advice. Our manuscript combined chiral SFG, AFM, and MD simulations to quantify the chirality transfer and magnified process in the supramolecular system, which draw the theoretical conclusions of the long-range chirality transfer from the intrinsic carbon atom to the nanorod supramolecular assembly. Since our focus in this manuscript is the L-GAn assemblies, we have not done the MD simulation for Fmoc-Glu-C18 assemblies. According to the reviewer's suggestion, we made an additional AFM experiment of the Fmoc-Glu-C18 monolayer and used these AFM images combined with the MD simulation from the L-GAn assembly to estimate the long-range chirality transfer of the Fmoc-Glu-C18 assembly.

The detailed discussion for the Fmoc-Glu-C18 assembly at the interface is highlighted below, which has been added to the main text and Supplementary Section 11. We also have added AFM measurements of the Fmoc-Glu-C18 monolayer in Fig. A1 (presented in Supplementary Fig. 15).

We added the following highlighted sentences in Supplementary Section 11.

As shown in Supplementary Figs. 15a-c, Fmoc-Glu-C18 molecules self-assemble at the air-water interface to form nanorod structures with a height of about 25 Å at a surface pressure of 15 mN/m. When the surface pressure is increased to 25 mN/m, the nanorods are more densely packed, and the defects of the monolayer formed by stacking

many nanorods are significantly reduced (Supplementary Figs. 15d).

We now attempt to characterize the Fmoc-Glu-C18 supramolecular self-assembly from AFM images using the same method as the L -GAN supramolecular self-assembly. We assume that L -GAN and Fmoc-Glu-C18 assemblies have similar three-dimensional single-molecule sizes at the interface because the molecular structures of Fmoc-Glu-C18 molecules and L -GAN molecules are similar (Fig. 1a), which of both self-assembly form nanorod structures with similar shapes (Supplementary Figs. 14a-b and Supplementary Fig. 15). Using the AFM results in Supplementary Fig. 15 and MD simulation of the L -GAN assemblies, we can roughly estimate the number of Fmoc-Glu-C18 molecules constructed a single nanorod. The Fmoc-Glu-C18 molecules at the interface are driven by π - π interaction to build the length of the nanorods along the y -direction of the coordinate axis and driven by the hydrogen bond to build the width of the nanorods along the x -direction of the coordinate axis. The length, width, and height of the nanorods can be estimated to be 400 molecules ($2187\text{\AA}/5.0\text{\AA}$, y -axis), 40 molecules ($375\text{\AA}/9.2\text{\AA}$, x -axis), and a single molecule (z -axis), respectively (Supplementary Fig. 15d). It must be pointed out that the number of Fmoc-Glu-C18 molecules assembled to form nanorods in the xyz direction above is just a rough estimate. In the future, we will calculate the length, width, and height of a single Fmoc-Glu-C18 molecule to form supramolecular assemblies at the interface by MD simulation to gain the exact number of molecules of the nanorods.

The main text has been changed accordingly and highlighted below.

We have changed the original sentence from “ L -GAN molecules self-assemble at the air/water interface to form nanofiber structures (Fig. 4a) where each nanofiber is assembled from multiple antiparallel β -sheet-like structures (Fig. 4b).” on page 11 in the main text, to “ L -GAN molecules self-assemble at the air/water interface to form nanorod structures (Supplementary Figs. 14a-b and Fig. 4a) where each nanorod is assembled from multiple antiparallel β -sheet-like structures (Fig. 4b).”

And we also modify the original sentence from “The steric hindrance of the strong intermolecular hydrogen of amide 2-3 and the twisting of the anthracene ring at the interface make the L -GAN molecules stacked horizontally and vertically to form

elongated nanofibers (Fig. 4c).” on page 11 in the main text, to “The L-GAn molecules stack along the x-axis to produce the width of the nanorods and along the y-axis to form the length of the nanorods due to the steric hindrance of the strong intermolecular hydrogen bond of amide 2-3 and the twisting of the anthracene ring at the interface. Here, we define the nanorods' length, width, and height along the y-axis, x-axis, and z-axis directions, respectively, as shown in Fig. 4c.”

And added the following sentence on page 12 in the main text “Compared with L-GAn molecules, the Fmoc-Glu-C18 molecule is connected to the glutamic acid unit through a flexible chain to produce steric hindrance, which may weaken the π - π stacking of aromatic rings and the intermolecular hydrogen bond, so that it forms shorter nanorod structures (Supplementary Fig. 15). The detailed quantitative number of molecules composing the Fmoc-Glu-C18 nanorods has been discussed in Supplementary S11.”

Fig. A1 AFM image of Fmoc-Glu-C18 monolayer. AFM images of one-layer Langmuir-Blodgett film of Fmoc-Glu-C18 deposited onto a freshly cleaved mica surface from the water subphase at (a-c) 15 mN/m and (d) 25 mN/m. The room

temperature is 25°C. The upper inset of Fig. A1(d) is an illustration of the nanorod in the three-dimensional coordinate direction. The length (2187 Å), width (375 Å), and height (25 Å) of the nanorod are composed of 400 molecules, 40 molecules, and single molecule, respectively.

2) Getting the theoretical results is further confirmed with experiential data making the conclusion more valid. Is it possible for the authors to investigate the CD spectra of relevant assembled systems to verify the chirality transfer results based on SFG-VS spectra?

Author reply: A valuable suggestion. In fact, the Liu group has studied the CD spectra of 40-layer L -/ D -GAN Langmuir-Schaefer films, as shown in Fig. A2, which proved the macroscopic chirality information transfer from the intrinsic chirality of the molecule to L -/ D -GAN supramolecular assemblies through chirality transfer. (Langmuir 2019, 35, 7, 2772-2779.). With the help of Liu's results, we further explain the reviewer's concerns as follows, and the change has been highlighted in the revised main text.

As shown in Fig. A2 from this literature, the UV-vis spectra of L -/ D -GAN in chloroform solution exhibit an intense absorption at 258 nm, which is attributed to the 1B_b absorption (Fig. A2 (a)). The 1B_b band of the Langmuir-Schaefer films of L -/ D -GAN shows a distinct blue-shifted with an absorption maximum at 250 nm, indicating H-aggregation of the anthracene groups in the film. Fig. A2 also shows four weak peaks at 347, 364, 384, and 396 nm in the 330-400 nm region, which are typical absorption peaks of anthracene that can be attributed to the 1L_a band as well as the S_0 - S_1 transition with different vibronic peaks (Fig. A2 (b)). Both the 1B_b - and 1L_a -bands of the anthracene of L -GAN and D -GAN show mirror-symmetric CD signals. These results confirmed our conclusion based on SFG-VS spectra that the anthracene rings form π - π stacking and the chiral information located at the chiral center is transferred to the anthracene rings. It is also worth pointing out that although CD spectra are a powerful tool to investigate the chirality of molecules in the bulk phase, they cannot study the chirality of the monolayer at the surface in situ due to the signal-to-noise ratio at the

surface and interfaces, as shown in Fig. A2. Therefore, chiral SFG-VS is probably one of the few promising spectroscopic tools to understand the intrinsic and structural chirality of molecular and supramolecular surfaces and interfaces.

The main text has been changed as below:

The original sentence on page 7 in the main text has been changed from “Furthermore, aromatic groups also generate chiral signals at 1621 cm^{-1} with mirror image peaks (Fig. 1g), indicating that the aromatic ring undergoes asymmetric packing induced by π - π interactions.” to “Furthermore, aromatic groups also generate chiral signals at 1621 cm^{-1} with mirror image peaks (Fig. 1g), indicating that the aromatic ring undergoes asymmetric packing induced by π - π interactions, which can also be confirmed by mirror-symmetric CD signals of the anthracene of the Langmuir-Schaefer films of L -GAn and D -GAn at 1B_b - and 1L_a -bands (Langmuir 2019, 35, 7, 2772-2779).”

Fig. A2 The CD spectra and UV-vis spectra of the Langmuir-Schaefer films of L - D -GAn. Circular dichroism (CD) spectra (top) and UV-vis spectra (bottom) of D -GAn (black line) and L -GAn (red line) 40-layer Langmuir-Schaefer films at 30 mN m^{-1} as well as L -GAn in chloroform (dotted line). (Langmuir 2019, 35, 7, 2772-2779.)

3) In Page 11, the authors determined the number of molecules on height (~ 30 molecules. I feel it should be height rather than thickness, which can distinguish from the approximate thickness of the monolayer), width, and length based on approximate thickness of monolayer from the width. However, one molecule usually exhibits different molecular lengths on the three dimensions, so the approximate thickness of monolayer

should not be used directly to determine the molecular number on the width and length. Therefore, the author should clearly show the method or explanation for the determination.

Author reply: We agree with the reviewer's comments on this issue. Indeed, our original statement in this manuscript was not very precise. We modify the original sentence in the main text's first paragraph on page 11 from “We determined the thickness of the L -GAN monolayer at equilibrium to be approximately 36 Å (Supplementary Fig. 10e), which is very similar to the 29-43 Å thickness of the nanofibers determined by AFM (Supplementary Fig. 11)⁴⁸. Moreover, we determined the width of the nanofiber, which is approximately one molecule; the thickness, which is ~30 molecules; and the length, which is ~900 molecules.” to “Here, we define the nanorods' length, width, and height along the y -axis, x -axis, and z -axis directions, respectively, as shown in Fig. 4c. We then determined the thickness of the L -GAN monolayer at equilibrium from MD simulation to be approximately 36 Å (Supplementary Fig. 13e), which is very close to about 29-43 Å height of nanorod in the z -axis direction from AFM³⁹ and is also approximately the height of one molecule. In the same way, we determined the width of the nanorod, about ~30 molecules along the x -axis; and the length, which is ~900 molecules along the y -axis, as shown in Fig. 4c in three dimensions.”

In addition, we have added more sentences highlighted below to Supplementary Section 11 to illustrate the detailed calculation to quantify the number of the molecules for one L -GAN self-assembly nanorod and added Fig. A3 as Supplementary Figs. 14 c-d.

We calculated the distance between the alkyl chain end methyl and the anthracene rings of the 100 L -GAN molecules that reached equilibrium after 55 ns MD simulation; the highest probability is located at 36 Å (Supplementary Fig. 13e). We deduced that the thickness of the monolayer is about 36 Å. The length and width of a single L -GAN molecule at the air/water interface are also calculated by MD simulation. Since molecules located at box boundaries tend not to have ordered orientations, we measured the length (9.2 Å) and width (5.0 Å) of several molecules in the central region

highlighted in Figs. 3b (bottom) and c (bottom) to estimate the molecular size of the self-assembly into nanorods accurately (Supplementary Figs. 14 c-d). We choose the longest nanorod measured by AFM (length 4560 Å and width 250 Å), and the _L-GAN molecules at the interface are driven by π - π interaction to build the length of the nanorods along the y-axis direction and driven by the hydrogen bond to build the width of the nanorods along the x-axis direction. The length, width, and height of the _L-GAN nanorods can be estimated to be 912 (4560 Å/5.0 Å, y-axis), 27 (250 Å/9.2 Å, x-axis) and single molecules (z-axis), respectively.

Fig. A3 The snapshots of _L-GAN molecules at 55 ns. (a) Front-view and (b) side-view of several _L-GAN molecules located in the central region of the box after 55 ns MD simulation.

4) *There are too many abbreviations in the manuscript, which may confuse readers. Some of them do not appear frequently, which is not necessary to use abbreviations. Please check and reorganize it as appropriate.*

Author reply: Thanks. In the revised manuscript, we have changed the abbreviation “BAM” in the main text to “**Brewster angle microscopy**” and changed the “cSFG” to “**chiral SFG**”; changed the “LB” to “**Langmuir-Blodgett**”; changed the “LS” to “**Langmuir-Schaefer**”.

5) *Regarding the organization of the Figures: a) In Figure 3, please adjust the arrows to avoid characters being hidden by dark red arrows. b) Figure 4 is too small for reading, please reorganize and amplify it.*

Author reply: Thanks for the referee's suggestion. We have adjusted the position of the arrows in Fig. 3 and enlarged Fig. 4 for easier reading, as shown below, and we have changed it accordingly in the manuscript.

6) About the TOC figure: The labels of the spectra are hidden by the graphical representation. It is not readable. Also, please reorganize the TOC scheme for better illustration of the significance of this paper.

Author reply: Good advice. In the revised manuscript, we reorganized the TOC scheme in the main text for a better illustration of the significance of our work, as shown below.

TOC. Chiral sum-frequency generation vibrational spectroscopy (chiral SFG-VS) was used to explore the molecular mechanism of chiral transfer at an air/water interface. The intrinsic chirality is transferred from the chiral center to the amide groups, anthracene ring, and alkyl chains. Subsequently, the chirality is transferred from the L-/D-GAn molecules to the self-assembled supramolecular assembly to form an antiparallel β -sheet-like structure.

7) The authors may consider citing the references related to diverse chiral self-assemblies, e.g., Adv. Mater. 2020, 32, 1801335; Chem. Sci. 2020, 11, 9989; ChemPhotoChem, 2022, 6, e202100256.

Author reply: Good advice. We have carefully read these relevant references and cited them in the main text to illustrate the potential applications of chiral supramolecular assemblies in preparing smart displays and optical data storage materials, which better

supports the significance of our research. In the revised manuscript, we have changed the sentence on page 3 from “Given the significance of symmetry breaking at interfaces, chiral supramolecular self-assembly at interfaces has been one of the most important research fields in recent years which are closely related to chiral living systems and are also proving to be an important tool for constructing large functional chiral complexes¹.” to “Given the significance of symmetry breaking at interfaces, chiral supramolecular self-assembly at interfaces has been one of the most important research fields in recent years, which are closely related to chiral living systems and is also proving to be an essential tool for constructing large functional chiral complexes¹ (Adv. Mater. 2020, 32, 1801335; Chem. Sci. 2020, 11, 9989; ChemPhotoChem, 2022, 6, e202100256).”

Reviewer #2:

The theory for this manuscript may need some further consideration.

1. SPP/PPS/PSP polarization SFG signals contain only chiral signals under the assumption that the molecules or self-assemblies are isotropic, i.e., $\chi_{XXX}=\chi_{YYY}=\chi_{XXY}=\chi_{XYX}=\chi_{YXX}=\chi_{YXY}=\chi_{YYX}=\chi_{XYY}=\chi_{YZZ}=\chi_{ZYZ}=\chi_{ZZY}=0$. Or a non-chiral system could have SPP/PPS/PSP signal as well. For example, some nonlinear crystals do not have chirality but have SPP/PPS/PSP signals, or please check Prof. Andre Knoesen's Biophysical Journal article in 2007 (table 1). This isotropic assumption is usually true for short chain SAMs samples but may not be true for systems with strong intermolecular interactions, since they can form crystals or microcrystals. For example, Prof. Andre Knoesen reported an example on Biophysical Journal in 2007 and Prof. Wei Xiong reported an example on JPCB in 2019. Even though the authors rotated the sample, the SFG signals generated at different time are not coherent and will not cancel out. Therefore, the measured effective second-order susceptibilities still contain anisotropic terms. If the sample is anisotropic, probably what the authors observed in this manuscript was not long-range chirality transfer, but was because these molecules formed a microcrystal as a whole. To check this, the authors should have measured SSS polarization signal. If SSS polarization signal is

non-zero, it means anisotropic interface and their data analysis needs to rewrite. If SSS polarization signal is zero, their analysis could be correct.

Author reply: Thanks for the valuable comments and suggestions. We have carefully evaluated the literature the referee suggested. We performed the additional sss polarization spectra at different surface pressures, as shown in Fig. A4, which confirms our hypothesis that the self-assemblies at the interface are isotropic. We have added Fig. A4 as Supplementary Fig. 1 and have added the highlighted sentences below in Supplementary Section 1.1.

As shown in Supplementary Figs. 1a-b, we performed sss polarization measurement of the $_D$ -GAn monolayer in the amide and C-H regions in the rotation sample cell. The results showed that the sss signal was zero (χ_{XXX} is zero), which confirmed that the $_L$ - $_D$ -GAn monolayer was an isotropic interface with C_∞ symmetry (Biophysical Journal, 2007, 93 (12), 4433-4444; J. Phys. Chem. B 2019, 123, 29, 6212-6221). In addition, we also measured the ssp and sss polarization spectra of the $_D$ -GAn monolayer at high surface pressure (Supplementary Fig. 1c) to further check the anisotropy of the interface. The results confirmed that no sss signal was detected neither due to experimental error nor low signal-to-noise ratio. Therefore, the $_L$ - $_D$ -GAn self-assemblies are isotropic at the interface and the self-assembly did not form the microcrystal.

It must be pointed out that we used a broadband SFG (resolution $\sim 8\text{ cm}^{-1}$) to do the additional sss polarization measurement to avoid the high laser intensity damage to the samples in the high-resolution sum-frequency generation (HR-SFG, resolution 0.4 cm^{-1}) we used.

The main text has been changed as below:

The original sentence on page 5 in the main text has been changed from “Here, we used the chiral SFG (Figs. 1b and c) combined with surface pressure, Brewster angle microscopy, and MD simulations to provide unique insights into the detailed mechanism of chirality transfer in the $_L$ - $_D$ -GAn supramolecular chiral self-assembled at the air/water interface at the molecular level.” to “Here, we used the chiral SFG (Figs. 1e and g) combined with the compression isotherms (π -A), Brewster angle microscopy and MD simulations to provide unique insights into the detailed mechanism of chirality

transfer in the L - D -GAN supramolecular chiral self-assembled at the isotropic air/water interface at the molecular level (Supplementary Fig. 1).”

Fig. A4 SFG spectral characteristics of D -GAN assemblies in the amide and C-H regions. Broadband sum-frequency generation vibrational spectra were recorded using sss and ssp polarization combinations for D -GAN monolayer in (a) amide region under

25 mN/m and C-H region under (b) 25 mN/m and (c) 40 mN/m at the air/water interface.

2. *The tilt angle calculation method used in this manuscript is under 2 assumptions:*

a. *The same assumption as in 1. If the system is anisotropic, more complicated analysis and calculation will be needed.*

b. *The assumption that the tilt angle distribution is narrow, i.e., $\langle \cos^3(\theta) \rangle / \langle \cos(\theta) \rangle = \langle \cos^2(\theta) \rangle$. As Prof. Garth Simpson and Prof. Kathy Rowlen pointed out on a JACS article in 1999, this assumption may not be true and could cause big trouble if the tilt angle is calculated to be around 39.2° , which is called magic angle. Therefore, Prof. Simpson and Prof. Rowlen claimed that if the calculated tilt angle is around 39.2° , especially $39 \pm 2^\circ$, special treatment would be needed. In this manuscript, the tilt angles were calculated to be 44° and 37° , while the authors did not provide special treatment in this manuscript.*

Author reply: Thanks for the reviewer's comments. As discussed above, the interface is isotropic based on the experimental results in Supplementary Fig. 1 (question 1 of reviewer 2). This excludes the limitation of anisotropy on the tilt angle calculation in the manuscript. We then carefully analyzed the tilt angle close to the "magic angle. We found that the magic angle should not affect our issue's main conclusions drawn from HR-SFG-VS orientational analysis. The detailed discussion is presented below.

We know that the orientation angle θ has always been calculated as follows. Assuming that the orientational distribution function is very narrow or simply a δ distribution, the apparent tilt angle θ can be calculated from the orientational parameter $D = \langle \cos \theta \rangle / \langle \cos^3 \theta \rangle$, which gets $\theta = \arccos(\sqrt{1/D})$. However, the validity of this calculation is considered to be problematic after the discovery of the SHG or SFG-VS magic angle by Prof. Garth Simpson and Prof. Kathy Rowlen. Magic angle is a precisely defined angle from the Legendre polynomial expansion of a Gaussian distribution in molecular orientation angle based on their assumption that δ -function

may not be justifiable in the calculation above. (J. Am. Chem. Soc. 1999, 121, 11, 2635-2636.). For SFG and SHG, the magic angle of $\theta = 39.2^\circ$ is obtained with $D = 5/3$. With the assumption that the orientation angle distribution function is a normalized Gaussian distribution, $f(\theta) = 1/(\sqrt{2\pi}\sigma) \exp(-(\theta - \theta_0)/2\sigma^2)$, a single D value is obviously not enough to provide information for two variables (i.e., the center angle θ_0 and the standard deviation σ). However, one crucial fact about the molecular interface is that the actual orientational angle distribution width of a given molecule on the surface or interface has been consistently shown to have limited values. Hongfei Wang et al., discussed this in-depth (Annu. Rev. Phys. Chem. 2015. 66:189-216), as shown in Fig. A5(a). They used the free O-H bond of water molecules at the topmost layer as an example to discuss the orientation angle of θ_0 and the standard deviation σ at the dynamic air/water interface. They found that with $\sigma < 15^\circ$, the allowed θ_0 values corresponding to the magic angle $\theta = 39.2^\circ$ can vary only less than 3° from $\sigma = 15^\circ$ to $\sigma = 0^\circ$. Therefore, the σ value of the O-H bond is limited, and the deviation of θ from θ_0 is significantly limited even in the magic angle region.

This is also true for the L-GAn monolayer constructed at the air-water interface. We used the ssp and ppp polarization intensity ratio to calculate the orientation angles of the CH₃ groups at different surface pressure, and we found the tilt angles were changed from 44° to 37° , which is very close to the magic angle. Even so, the magic angle should not affect the main conclusions drawn from HR-SFG-VS orientational analysis in our issue. The reasons are as follows. Firstly, the narrow spectral feature centered at 2880 cm^{-1} (FWHM is about 6.8 cm^{-1} , as shown in Supplementary Table 6) suggests that the CH₃ group is quite ordered at the interface, which indicates the σ value of the methyl groups at the interface is limited. Secondly, the change of the D value of the CH₃ group at two surface pressure is also an indication that the σ value of the CH₃ group is limited. If the σ value is broad, the D value would not change. Third, as the analysis in the literature for the free O-H bond of water molecules at the topmost layer suggests, even if one assumes the relatively large σ as 15° , the allowed θ_0 values corresponding to the magic angle $\theta = 39.2^\circ$ can vary only less than 3° from $\sigma = 15^\circ$ to $\sigma = 0^\circ$. For the

amphiphilic molecule at the interface, the dynamic of the CH₃ group located at the long alky chain is not fast as the air/water interface, and the standard deviation σ of CH₃ should be smaller than free OH.

Moreover, as shown in Fig. A5(b), we use the mathematic simulated ssp/ppp ratio for the CH₃-SS peak to calculate the orientational tilt angle. The horizontal dash line indicates the ratio from experimental fittings, and the vertical dashed lines give the range of the orientational angle of the CH₃ group with the gaussian distribution of $\sigma = 15^\circ$ and $\sigma = 0^\circ$. With $\sigma < 15^\circ$, the allowed θ_0 values corresponding to the tilt angle $\theta = 37^\circ$ can vary from 34 to 37, that is only less than 3° from $\sigma = 15^\circ$ to $\sigma = 0^\circ$. The tilt angle $\theta = 44^\circ$ can vary from 43 to 44, that is only less than 1° from $\sigma = 15^\circ$ to $\sigma = 0^\circ$. Therefore, the existence of the SFG-VS magic angle should not affect the main conclusions drawn from our HR-SFG-VS orientational analysis, the spectra provided the orientational parameter D was determined with enough accuracy.

We have replaced Supplementary Fig. 12 with Fig. A5(b) and added the following sentences in Supplementary Section 9:

For the L-GAn monolayer constructed at the air-water interface, we used the ssp and ppp polarization intensity ratio to calculate the orientation angles of the CH₃ groups at different surface pressure and we found the tilt angle was changed from 44° to 37° which is very close to the magic angle (Supplementary Fig. 12). But this magic angle should not affect the main conclusions drawn from HR-SFG-VS (0.4 cm^{-1}) orientational analysis. The reasons are as follows. Firstly, the narrow spectral feature centered at 2880 cm^{-1} (FWHM is 8.6 cm^{-1} , as shown in Supplementary Table 6) suggests that the CH₃ group is quite ordered at the interface, which indicates the σ value of the methyl groups at the interface is limited. Secondly, the change of the D value of the CH₃ group at two surface pressure are also an indication that the σ value of the CH₃ group is limited. If the σ value is broad the D value would not change. Third, as the analysis in the literature for free O-H bond of water molecules at the topmost layer suggest, even one assume the relatively large σ as 15° , the allowed θ_0 values corresponding to the magic angle $\theta = 39.2^\circ$ can vary only less than 3° from $\sigma = 15^\circ$ to $\sigma = 0^\circ$ (Annu. Rev. Phys. Chem. 2015. 66:189-216). For the amphiphilic molecule at the interface, the dynamic

of the CH₃ group located at the long alky chain are not fast like air/water interface, and the standard deviation σ of CH₃ should be smaller than the free OH.

In Supplementary Fig. 12, we use the mathematic simulated ssp/ppp ratio for the CH₃-SS peak to calculate the orientational tilt angle. The horizontal dash line indicates the ratio from experimental fittings, and the vertical dashed lines give the range of the orientational angle of the CH₃- group with the Gaussian distribution of $\sigma = 15^\circ$ and $\sigma = 0^\circ$. With $\sigma < 15^\circ$, the allowed θ_0 values corresponding to the tilt angle $\theta = 37^\circ$ can vary from 34 to 37°, that is only about 3° from $\sigma = 15^\circ$ to $\sigma = 0^\circ$. The tilt angle $\theta = 44^\circ$ can vary from 43 to 44, that is only about 1° from $\sigma = 15^\circ$ to $\sigma = 0^\circ$. Therefore, the existence of the SFG-VS magic angle should not affect the main conclusions drawn from our HR-SFG-VS orientational analysis, the spectra provided the orientational parameter D was determined with enough accuracy.

Figure A5. The apparent molecular tilt angle (calculated by incorrectly assuming

a δ -function distribution, as in eq $D \equiv \frac{\langle \cos^3 \theta \rangle}{\langle \cos \theta \rangle} = \frac{\chi_{ZZZ}^{(2)}}{\chi_{ZZZ}^{(2)} + 2\chi_{ZXX}^{(2)}} \cong \cos^2 \langle \theta \rangle$) as a

function of the root-mean-square width (σ) of a Gaussian distribution. (a)

Illustration of the second harmonic generation (SHG) and sum-frequency generation vibrational spectroscopy (SFG-VS) magic angle and how the physical constraint of the distribution width can significantly recover the confidence in SFG-VS orientational analysis. (Annu. Rev. Phys. Chem. 2015. 66:189-216.) (b) The ratio of symmetric stretching vibration intensity under ssp and ppp polarization for CH₃ group vs the

orientation angle θ under δ distribution and Gaussian distribution ($\sigma=10^\circ$ and $\sigma=15^\circ$). The purple line is under δ distribution. The red and blue lines are under Gaussian distribution. The horizontal dash line indicates the ratio from experimental fittings, the vertical dash line indicates orientation angle of the CH_3 group at different orientation distribution.

3. The author claimed “In addition, we also found (Supplementary Fig. 6a) that the ssp intensity in the amide region of the L-D-GAn monolayer increases as the surface pressure increases, indicating the amide groups of L-D-GAn become more ordered at higher surface pressures, which is always true for other monolayers”. However, when monolayer surface pressure increases, the surface area (per molecule) should decrease, which indicates a higher molecular density per area. Therefore, I wonder if the authors considered the relationship between SFG signal intensity vs. molecular density.

Author reply: Thanks for the referee's question. We have added the following sentences in Supplementary Section 6: According to the SFG theory in S16 ($\chi_{IJK,q}^{(2)} = N \sum_{i,j,k} \langle R_{Ii} R_{Jj} R_{Kk} \rangle \beta_{ijk,q}^{(2)}$), the SFG intensity of any SFG experiment is proportional to the number density of the interface moiety, the orientation angle, and the distribution function. The enhancement of the SFG signals with the surface pressure increases is based on the resultant contribution of the increase in molecular number density and the change in molecular conformation when the surface pressure increases. As reported in the literature (J. Phys. Chem. C 2009, 113, 4088-4098.), the increased ratio of symmetric stretching vibration intensity of methyl and methylene groups indicates the increase in relative ordering and increases the detected SFG signal (Supplementary Fig. 8b). Such an increased ordering effect for the alkyl chains indicates the L-D-GAn molecules are more ordered at the higher surface pressure.

Moreover, by comparing the χ_{ssp}/χ_{sps} ($A_{q,ssp}/A_{q,sps}$) values of amide I of amide 1 (Supplementary Tables 1-2) for amide groups of L-D-GAn, we obtained an intensity

ratio $\left(\frac{\chi_{ssp,amide\ I\ of\ amide\ 1}^{(2)}}{\chi_{sps,amide\ I\ of\ amide\ 1}^{(2)}}\right)$ from -0.50 to -0.37 for _D-GAn and from -0.50 to -0.41 for _L-

GAn with the surface pressure increasing. Changes in the intensity ratio of the ssp and sps spectra of the amide I band indicate a change in the conformation of the amide group, which also caused the change in the signal intensity. Therefore, the increase of the SFG signal intensity in the amide and the C-H bands with the surface pressure increases is due to the combined contribution of the increase in molecular number density and the change in molecular conformation.

Reviewer #3:

The authors study a monolayer of supramolecular self-assemblies of a chiral amphiphilic molecule at the air/water interface. They use methods of chiral sum frequency generation (SFG) vibrational spectroscopy and molecular dynamics (MD) simulation. The chiral amphiphilic molecule has two C18 carbon chains and one aromatic anthracene functional group. These three groups are connected to a small linker backbone via three amide groups. One of the amide groups that links to anthracene is connected to a carbon atom that is a chiral center. The authors took chiral SFG spectra of the enantiomers of this molecule at the air-water interface. The SFG spectra show that the carbon chains of the two enantiomers give the same chiral SFG response for the methyl and methylene stretching modes but with opposite signs. Moreover, the authors also assign the vibrational C=C stretching modes of anthracene and observe these modes are also with opposite phase for the two enantiomers. The authors therefore conclude that chirality is transferred from the chiral-center carbon atom to the two carbon chains and to the anthracene group. This major conclusion is supported by experiments. Because the long-range chirality transfer is of interest to many research fields, the manuscript should be accepted for publication if the authors can fully address the following major questions and concerns:

Major concerns:

1. The major observations to support the main point is that the enantiomers give the chiral SFG response of the methyl and methylene stretches with opposite phase. However, in the manuscript, the authors first discuss the interpretation and assignments of amide I and amide II modes in order to assignment the C=C stretches of anthracene. Although these assignments need to be done, it is not crucial to the main theme of the paper – chirality transfer from the chiral-center C atom via long distance (2-3 nm) to the methyl and methylene groups on the carbon chains. Thus, the authors may want to

reduce the emphasis of the assignments and interpretations of the amide spectra.

Author reply: Thanks for the suggestion. We have simplified the assignment of the three amide groups in this section to focus on the chirality transfer. However, we still keep the assignments and interpretations of the amide spectra because the information for amide I and amide II modes are essential to understand the interaction among the three amide groups and how the secondary structure of the self-assemblies affects the shape of the supramolecular self-assemblies, from which we can quantify the long-range chirality transfer in the supramolecular self-assemblies.

The revision is highlighted in the second paragraph on page 5.

The original sentence on page 5 in the main text has been changed from " By comparing the peaks of D -GAN and Fmoc-Glu-C18 in the amide band (Figs. 1d and f, and Supplementary Figs. 4b and c), the peaks at 1657 and 1598 cm^{-1} are assigned to the amide I and II bands of amide 1 of D -GAN. In the same way, by comparing the spectra of the Fmoc-Asp-C17 and Fmoc-Glu-C18 monolayers (Supplementary Figs. 3b and c), the peaks at 1653 and 1568 cm^{-1} are assigned to the amide I and II bands of amide 2 of Fmoc-Asp-C17, and the peaks at 1647 and 1555 cm^{-1} are assigned to the amide I and II bands of amide 2 of Fmoc-Glu-C18, respectively. The other two amide peaks of D -GAN at 1637 and 1578 cm^{-1} are assigned to the amide I and amide II bands of amide 3." to "The shifted peak positions were assigned to the amide I and II bands of the amide groups due to the different chemical environments. Through two control experiments, we can distinguish the amide I and II bands of the three amide groups of the glutamic acid group. The results are shown in Fig. 1d and Table 1, and the details of spectral peak assignments are written in Supplementary Section 4."

We also have changed sentences in Supplementary Section 4 from "In summary, through two sets of controlled experiments, we can accurately assign the signal peaks from the three amide groups of the glutamate head group of D -GAN detected in the SFG spectra." to "In summary, by comparing the peaks of D -GAN and Fmoc-Glu-C18 in the amide band (Figs. 1d and f, and Supplementary Figs. 5b and c), the peaks at 1657 and 1598 cm^{-1} are assigned to the amide I and II bands of amide 1 of D -GAN. In the same way, by comparing the spectra of the Fmoc-Asp-C17 and Fmoc-Glu-C18 monolayers

(Supplementary Figs. 4b and c), the peaks at 1653 and 1568 cm^{-1} are assigned to the amide I and II bands of amide 2 of Fmoc-Asp-C17, and the peaks at 1647 and 1555 cm^{-1} are assigned to the amide I and II bands of amide 2 of Fmoc-Glu-C18, respectively. The other two amide peaks of D-GAn at 1637 and 1578 cm^{-1} are assigned to the amide I and amide II bands of amide 3.”

2. In interpreting the amide spectra, the authors discuss the data in the context of protein secondary structures. This can be another distraction. In this study, although the molecule under study contains three amide functional groups, they are not connected in the same way as standard native peptide bonds in a protein. Hence, the molecule is not likely form standard protein secondary structures by itself or via self-assembling, especially that the conformation of the molecule should be constrained by the amphiphilic air-water surface. Therefore, it can be problematic to discuss how the molecule can form beta-sheet and alpha-helical structures. Indeed, if one wants to make the points about forming these types of secondary structures, analysis of Ramachandran angles is typically needed.

Author reply: We appreciate the referee's advice. We note that the secondary structure we used in the original manuscript may lead to some misunderstandings. In previous reports, chiral SFG has been used to study the secondary structures of the peptide and proteins (J. Am. Chem. Soc. 2010, 132, 15, 5405-5412; J. Am. Chem. Soc. 2011, 133, 21, 8094-8097; J. Phys. Chem. B 2015, 119, 7, 2769-2785.). We here use chiral SFG to study the chiral supramolecular assemblies and we found the chiral SFG peak position of the L-/D-GAn amphiphilic molecules at the interface are similar to the peptides and proteins at the interface. Since L-/D-GAn molecules are not connected in the same way as standard native peptide bonds, for clarity, we redefine and calculate the Ramachandran angles in chiral supramolecular assembly systems by following the definition of Ramachandran angles in peptide and protein systems. Moreover, we use antiparallel **β -sheet-like** structures to demonstrate the formation of the chiral structure of L-/D-GAn monolayer. The detailed discussion is highlighted below, and we added

them to the main text and the Supplementary Section 5.

We changed “ **β -sheet**” to “ **β -sheet-like**” in the main text and added the sentence “Because L/D -GAN molecules are not connected in the same way as standard native peptide bonds, for clarity, we redefine and calculate the Ramachandran angles in chiral supramolecular assembly systems by following the definition of Ramachandran angles in peptide and protein systems, as shown in Supplementary Fig. 7, which confirms the results of chiral SFG spectroscopy that the L/D -GAN assembly forms an antiparallel β -sheet-like structure.” in the main text on page 6.

We have added Fig. A6 as Supplementary Fig. 7 and the following sentences in Supplementary Section 5.

As we reported, amide 2 and amide 3 of the L -GAN molecule have the strongest ability to form inter/intra-molecular hydrogen bonds (Fig. 3d). As a result, it formed an antiparallel β -sheet-like structure, which was also confirmed by polarization analysis of the spectral peaks in the amide band and symmetry theory (Supplementary Section 5). Following the Ramachandran definition (J. Mol. Biol, 1963, 7, 95-99.), we can propose a secondary structure-like conformational in the supramolecular system. We first define the dihedral angles Φ and Ψ of the plane [CO-NH] where amide 2 and amide 3 are located, respectively (Supplementary Fig. 7a). We then performed MD simulation to calculate the molecular conformation of the L -GAN molecule under an equilibrium state (55 ns). For ease of visualization and data analysis, only amide 2 and amide 3 groups and hydrogen bonds formed by amide 2 and amide 3 groups are shown (Supplementary Fig. 7b). We finally counted the dihedral angles of the amide 2 and amide 3 groups formed 2-3 hydrogen bonds, respectively, and plotted the scatter density chart as shown in Supplementary Fig. 7c. In Supplementary Fig. 7c, the deep color represents the area where the dihedral angles Φ and Ψ are mainly distributed. The dihedral angles commonly found in parallel ($\Phi = -119^\circ$, $\Psi = 113^\circ$) and antiparallel ($\Phi = -139^\circ$, $\Psi = 135^\circ$) β -sheets which fall in the upper left-hand quadrant of a Ramachandran plot (Chem. Rev. 2010, 110 (6), 32-69.). It is clearly noticed that the dihedral angles for amide 2 and amide 3 are mainly located in the β -sheet region of the Ramachandran plot (J. Phys. Chem. B 2019, 123, 5769-5781; J. Mol. Biol. 2002, 317

(2), 291-308; J. Phys. Chem. B 2021, 125 (17), 4274-4285). Combining the SFG spectra, we infer that L-GAn molecules form an antiparallel β -sheet-like structure at the air-water interface. It is worth mentioning that since molecules are difficult to self-assemble into standard protein secondary structures, the L-GAn molecules form an approximate secondary structure through self-assembly rather than the standard protein secondary structure.

Fig. A6 Ramachandran angles of the antiparallel β -sheet-like structure formed by L-GAn supramolecular assemblies. (a) The twist angles of the plane [CO-NH] where L-GAn molecules amide 2 (Φ) and amide 3 (Ψ) are located, respectively. (b) Top-view snapshot of last frame (55 ns). Only amide 2 and amide 3 groups and hydrogen bonds formed by amide 2 and amide 3 groups are shown. (c) Scatter density chart of the dihedral angle Φ of amide 2 and the dihedral angle Ψ of amide 3 that form 2-3 hydrogen bonds.

3. It is like that the authors spent a lot of effort to interpret the amide spectra because they need to assign the anthracene C=C stretches. The frequency of the aromatic C=C

stretches falls into the amide I region. Thus, to solidly show chirality transfer from the chiral-center carbon to the anthracene group, the authors need to not only assign the C=C stretches but also all the other amide bands. Nonetheless, the authors can also consider probing the aromatic C-H stretches of the anthracene group, which should be at frequency > 3000 cm⁻¹. Perets et al. (Langmuir, 2022) showed recently that aromatic bases in DNA show chiral C-H stretches at frequency > 3000 cm⁻¹ upon hybridization. If the anthracene groups also form chiral assemblies, they may also show chiral SFG response of aromatic C-H stretches.

Author reply: Good questions. In our original manuscript, we conducted the SFG spectra ranging from 2750 to 3150 cm⁻¹, which covered the =C-H stretching vibration band of the anthracene group. However, in this measurement, both the chiral SFG signal and achiral SFG signal of the =C-H stretching vibrations of the anthracene groups of L-GAn and D-GAn molecules are not observed (Figs. A7 (c) and (e)). Therefore, we now perform additional SHG detection (Figs. A7 (d) and (f)) and combine it with SFG measurement in the amide band, MD simulation, and CD detection to further discuss the =C-H peaks.

In this revised manuscript, we have highlighted the changed sentences in the main text and added Figs. A7 as Supplementary Figs. 10 and highlighted the modified sentences in Supplementary Section 8 as follows.

We add the following sentences in the main text on page 8.

“However, neither the chiral SFG signal nor the achiral SFG signal of the =C-H stretching vibration (central peak at 3046 cm⁻¹) of the anthracene group of the L-GAn and D-GAn molecules could be observed. The reasons are discussed in Supplementary Section 8.”

We added the following sentences in Supplementary Section 8.

As shown in the infrared spectrum of the anthracene molecule, we have observed the =C-H stretching vibration peak of the anthracene ring at 3046 cm⁻¹ (Supplementary Fig. 10a). However, in the SFG measurement at 2750-3150 cm⁻¹, both the chiral SFG signal and achiral SFG signal of the =C-H stretching vibrations of the anthracene groups of L-GAn and D-GAn molecules (center peak at 3046 cm⁻¹) cannot be observed

(Supplementary Figs. 10c and e) even the IR laser intensity at $3000\text{-}3100\text{ cm}^{-1}$ is high enough (Supplementary Fig. 10b). Supplementary Fig. 10b shows the non-resonant SFG signal from z-cut quartz, from which we conclude that the IR laser intensity is high enough to get reasonable SFG spectra for =C-H stretching vibrations of the anthracene groups. The absence of the =C-H SFG signal might have two reasons. One is that the anthracene ring has a centrosymmetric structure that the dipole moment of the =C-H has been canceled out. The second reason is the anthracene rings parallel to the interface; which can be confirmed by MD simulation (about 70° with an orientation contribution of $0\text{-}160^\circ$ (Fig. 3f).) Moreover, the weak resonant SHG signal on the D -GAN monolayer which indicated the anthracene ring should lie on the interface because the typical absorption of anthracene is in the $330\text{-}400\text{ nm}$ when the incident laser beam is 780 nm (Supplementary Fig. 10d).

It should be noted that although the chiral SFG response of aromatic =C-H stretches is not present, we can still conclude that the chirality transfer from the chiral-center carbon to the anthracene group from CD spectra and SHG spectra. The CD spectra of 40-layer Langmuir-Schaefer films of L -GAN and D -GAN show both the 1B_b - and 1L_a -bands of the anthracene of L -GAN and D -GAN show mirror-symmetric CD signals (Langmuir 2019, 35, 7, 2772-2779.). And the chiral SHG spectra of the D -GAN and L -GAN monolayer at the air/water interface shows opposite chiral signals (Supplementary Fig. 10f). Moreover, we also detected mirror image chiral SFG signals of C=C stretching vibration peaks in the anthracene groups of L -GAN and D -GAN molecules in the amide band (Figs. 1e and g). These results indicate that during the interfacial assembly of L -GAN and D -GAN molecules, their aromatic groups exhibit opposite chiral characteristics by asymmetric stacking and twisting induced by chiral centers.

Fig. A7 Aromatic =C-H stretching vibration spectra of anthracene groups of D-GAn molecules. (a) Fourier transform infrared spectroscopy (FTIR) of anthracene molecules (KBr pellet method). (b) SFG spectra of the z-cut quartz under ssp polarization. (c) Achiral SFG spectra of the D-GAn monolayer at a surface pressure of 40 mN/m in the C-H band at the air/water interface. (d) P-polarization detection of the 390 nm SHG signal from the D-GAn monolayer and pure H₂O at the air/water interface. (e) Chiral SFG spectra of the L-GAn and D-GAn monolayers at different surface pressures at the air/water interface. (f) The s-polarization detection of the 390 nm SHG signal from the D-GAn and L-GAn monolayer at the air/water interface ($DCE = \frac{\Delta I}{I} = \frac{2(I_{+45^\circ s} - I_{+135^\circ s})}{(I_{+45^\circ s} + I_{+135^\circ s})}$, the DCE values of the D-GAn and L-GAn monolayer are -0.02 and 0.03, respectively).

4. The authors propose a structural model of supramolecular assemblies (Figure 4). In the model, the chiral amphiphilic molecules form beta-sheet structures. For the reason discussed in point#2, it is unclear how the beta-sheet structure is formed. If the authors want to make the claim, they should at least build a molecular model that form beta-sheet structures using MD and then analyze the Ramachandran angles. This additional evidence is necessary because the amphiphilic molecule can have orientational constraints at the interface and these orientational constraints can hinder formation of beta-sheet structures.

Author reply: Thanks for the valuable question. We supplemented the Ramachandran angles analysis to confirm the formation of the antiparallel β -sheet-like structure; please see the answer to question 3 (Fig. A6).

5. Although the chirality transfer reported in the current study is impressive, the authors need to provide additional backgrounds in the Introduction about long-range chirality transfer. Is it the first observation of chirality transfer over 2-3 nm? Some background information may help highlight the novelty and significance of the work that can justify publication in *Nature Communications*.

Author reply: Thanks for your valuable comments. On page 3 of the revised main text, we have added the following sentence after the sentence "The mechanism of the development of supramolecular chirality has always been a fascinating scientific topic in this field. " to illustrate the importance of chirality transfer better: "It has been found that in living cells, the intrinsic chirality of cells induces the directional rotation of tissue during the formation of tissues, which is a potential mechanism for the morphogenesis of left-right asymmetric tissue (Science, 2011, 333 (6040), 339-341.). However, how intrinsic chirality induces asymmetric stacking during assembly and how far the chirality transfers from the intrinsic carbon to the self-assemblies remains unclear."

And we have modified the original sentence at the end of the first paragraph on page 4 of the main text from "We found that there is a long-range chirality transfer in the system." to "We found that in supramolecular assemblies, chiral information from

chiral centers can be transferred to hundreds of molecules within 400-500 nm distance between the molecules through weak non-covalent interactions. As far as we know, quantitative such long-range chirality transfer is very rare in constructing supramolecular systems from small molecules.”

Other suggestions:

a. Page 4: “*inter/Intra hydrogen bonds.*” The term “*intra*” should be in lower case.

Author reply: We thank the referee for the suggestion. We modify the original sentence from “It has two carboxylic groups and one amino group, which can form three amide groups and then form inter/Intra hydrogen bonds.” to “It has two carboxylic groups and one amino group, which can form three amide groups and then form inter/intra-molecular hydrogen bonds.”

b. Page 4: “*pi-A.*” The authors may want to define *pi* and *A* for readers who are not familiar with surface pressure measurements.

Author reply: Thanks for your valuable comments. In order to make our manuscript easier for readers to understand, we add more sentences in Supplementary Section 2: The phase behavior of _L-GAN at the air/water interface was investigated by surface pressure measurements (isotherm experiments) and Brewster angle microscopy. The surface pressure (π) is defined as the interfacial tension difference between a clean water interface (γ_0) and an interface in the presence of the amphiphilic molecule (γ); the expression is $\pi = \gamma_0 - \gamma$. _L-GAN molecules are spread on the water surface and compressed by the barriers. The surface pressure (π) is plotted as a function of the area (*A*) per molecule at a constant temperature. The π -*A* isotherms can reflect phase states, such as gas (G), liquid (L), and solid (S) and various mesophases between ideal liquid and solid states at the air-water interface during compression.

c. Page 9: “*Fig.3a*” and “*Fig. 3d*” should be “*Figure 3a*” and “*Figure 3d.*”

Author reply: Thanks for the referee’s suggestion. We checked the writing format for

articles recently published in Nature Communication, then we used Fig. * to number the figures.

d. Figure 2b: “s+/-mp” should be written out as the actual physical quantity being measured, i.e., different between (s + mp) and (s – mp). The current label could be confusing especially to readers who are not in the SFG field.

Author reply: We appreciate the referee’s advice. We have modified the sign of the interference chiral polarization in the main text in Fig. 1g and Fig. 2b from “s±mp” to “s(+m)p-s(-m)p”. Fig. 1g and Fig. 2b have been updated in the revised manuscript.

REVIEWER COMMENTS

Reviewer #1 (Remarks to the Author):

I have looked over the authors' point-by-point response and revisions. I think the authors have well addressed my comments. The revised version is now in a good shape. I would recommend its acceptance.

Reviewer #2 (Remarks to the Author):

The response convinced the reviewer. It would be better if the manuscript main text can be edited accordingly.

1. For the sss experiment, one or two simple sentences can be added into the manuscript, specifically saying sss polarization is negligible and guide the readers to supplementary for more details.
2. They can add one simple sentence into the manuscript saying their analysis conclusion would not be affected by magic angle, and guide the readers to supplementary for more details.
3. They can cite supplementary after this sentence "In addition, we also found (Supplementary Fig. 8a) that the ssp intensity in the amide region of the L-/D-GAnmonolayer increases as the surface pressure increases, indicating the amide groups of L-/D-GAn become more ordered at higher surface pressures" to guide readers to supplementary for more details.

Some other things that need attention:

1. All the tables in supplementary look weird, probably due to format issue.
2. There is a typo in manuscript page 19 line 366. In the equation, it should be proportion to symbol rather than infinity symbol.
3. Just curious about what is the visible pulse profile in time domain. Gaussian profile or etalon profile?

Reviewer #3 (Remarks to the Author):

The authors made a mistake in defining the Ramachandran angles (Fig. S7), and therefore the analysis is not valid.

It is suggested that the authors refer to any biochemistry textbook or discuss with biochemists about definition of Ramachandran angles before redoing the analysis. Consequently, the discussion about secondary structures may need to be modified.

Response to Reviewers

We thank the Reviewers for carefully reading the manuscript and for the valuable suggestions. In the revision of this manuscript, we took these suggestions into full consideration. Moreover, we changed and supplemented some of the discussions in this manuscript as suggested by reviewer #3. The revisions in this manuscript has been highlighted in the response letter. At the same time, the revised main text and the supporting information has also been highlighted. We hope this revision will satisfy reviewer #3's concerns.

Below, we give a point-by-point response to the comments of the referees.

Reviewer #1:

I have looked over the authors' point-by-point response and revisions. I think the authors have well addressed my comments. The revised version is now in a good shape. I would recommend its acceptance.

Reviewer #2:

The response convinced the reviewer. It would be better if the manuscript main text can be edited accordingly.

Author reply: Thanks for the nice suggestions. We have changed our manuscript and SI according to the reviewer's advice.

1. For the sss experiment, one or two simple sentences can be added into the manuscript, specifically saying sss polarization is negligible and guide the readers to supplementary for more details.

Author reply: Thanks for the valuable comments and suggestions. We have revised the following sentences in the main text in the first paragraph on page 5.

The original sentence on page 5 in the main text has been changed from "Here, we used the chiral SFG (Figs. 1e and g) combined with the compression isotherms (π -A), Brewster angle microscopy and MD simulations to provide unique insights into the detailed mechanism of chirality transfer in the L-/D-GAn supramolecular chiral self-assembled at the isotropic air/water interface at the molecular level (Supplementary Fig. 1)." to "Here, we used the chiral SFG (Figs. 1e and g) combined with the compression isotherms (π -A), Brewster angle microscopy and MD simulations to provide unique

insights into the detailed mechanism of chirality transfer in the L-/D-GAn supramolecular chiral self-assembled at the isotropic air/water interface (the silent sss polarization proves that the interface is isotropic at the molecular level, see Supplementary Fig. 1 for details).”

2. *They can add one simple sentence into the manuscript saying their analysis conclusion would not be affected by magic angle, and guide the readers to supplementary for more details.*

Author reply: Thanks for the suggestion. The revision is highlighted in the first paragraph on page 11 of the main text.

The original sentence on page 11 in the main text has been changed from “Fig. 3e shows that the orientation angles of the two long alkyl chains mainly fall within approximately 40°, which is consistent with the orientation angle in the SFG analysis (Supplementary Section 9) in the experiment above.” to “Fig. 3e shows that the orientation angles of the two long alkyl chains mainly fall within approximately 40°, which is consistent with the orientation angles of the alkyl chains. In addition, we confirmed that this conclusion from SFG analysis in the above experiments would not be affected by magic angle (Supplementary Section 9).”.

3. *They can cite supplementary after this sentence "In addition, we also found (Supplementary Fig. 8a) that the ssp intensity in the amide region of the L-/D-GAn monolayer increases as the surface pressure increases, indicating the amide groups of L-/D-GAn become more ordered at higher surface pressures" to guide readers to supplementary for more details.*

Author reply: Thanks for the valuable suggestion. This revision is highlighted in the first paragraph on page 8 of the main text.

The original sentence on page 8 in the main text has been changed from " In addition, we also found (Supplementary Fig. 8a) that the ssp intensity in the amide region of the L-/D-GAn monolayer increases as the surface pressure increases, indicating the amide groups of L-/D-GAn become more ordered at higher surface pressures, which is always true for other monolayers⁴⁰." to “In addition, we also found (Supplementary Fig. 8a)

that the ssp intensity in the amide region of the L - D -GAn monolayer increases as the surface pressure increases, indicating the amide groups of L - D -GAn become more ordered at higher surface pressures, which is always true for other monolayers⁴⁰ (In Supplementary Section 6 for details).”.

Some other things that need attention:

1. *All the tables in supplementary look weird, probably due to format issue.*

Author reply: Thanks for the reviewer's comment. The supplementary table format has been slightly reformatted for ease of viewing.

2. *There is a typo in manuscript page 19 line 366. In the equation, it should be proportion to symbol rather than infinity symbol.*

Author reply: We appreciate the referee's advice. We have corrected the infinity symbol in the equation on page 19 of the original manuscript to a proportional symbol.

3. *Just curious about what is the visible pulse profile in time domain. Gaussian profile or etalon profile?*

Author reply: The visible pulse profile is Gaussian profile in the time domain. We have added this sentence in the experiment section of the manuscript on page 20.

Reviewer #3 (Remarks to the Author):

The authors made a mistake in defining the Ramachandran angles (Fig. S7), and therefore the analysis is not valid.

It is suggested that the authors refer to any biochemistry textbook or discuss with biochemists about definition of Ramachandran angles before redoing the analysis. Consequently, the discussion about secondary structures may need to be modified.

Author reply: Thanks for your thoughtful suggestion. We appreciate and agree with the reviewer's comments. After carefully checking the relevant literature and biochemistry textbook, we agree that the secondary structures used in our original manuscript might be confusion to the reader with the secondary structures of proteins in biochemistry. Since the structures of the self-assemblies at the interface do not exactly follow the definition of antiparallel β -sheet structure in protein and polypeptides (Figures. B1 (a)), we no longer use Ramachandran angle to describe the antiparallel β -sheet-like structure of the L - D -GAn assembly in this revised manuscript. The reasons

for this will be detailed below. At the same time, we remove original Supplementary Figure 7 and related descriptions in Supplementary Section 5 in revised manuscript. The secondary structure is no longer used in the revised manuscript and SI, instead, we used β -sheet-like structure directly in the manuscript. The reasons for using β -sheet-like are discussed below and the main points of these discussions are supplemented in the main text and SI.

According to the *chapter 4 of Lehninger Principles of Biochemistry* (7th edition), the secondary structure refers to any chosen segment of the polypeptide chain and describes the local spatial arrangement of its main-chain atoms, without regard to the positioning of its side chains or its relationship to other segments. For proteins, every type of secondary structure can be completely described by the dihedral angles (*Ramachandran angles*) ϕ and ψ associated with each residue. The α carbons of adjacent amino acid residues in protein are separated by three covalent bonds, arranged as C_α -C-N- C_α . The N- C_α and C_α -C bonds can be rotated and are described by dihedral angles ϕ and ψ , respectively (Figure. B1 (a)). A regular secondary structure occurs when each dihedral angle, ϕ and ψ remains the same or nearly the same throughout the segment. For the antiparallel β -sheet structure, the adjacent polypeptide chains have the opposite amino-to-carboxyl orientations. The repeat period is 7 Å and the interstrand hydrogen bonds are essentially in-line (Figure. B1 (b)). Dihedral angles ϕ and ψ are suitable for describing the stable secondary structure of proteins with a cyclic side-chain topology linked by standard peptide bonds.

However, the L -/ D -GAN molecules are consist of three amino acid components, which is difficult to form standard secondary structures such as β -sheet or α -helix by self-assembly. Because the intermolecular hydrogen bonds among three amide groups of the L -/ D -GAN molecule are linked differently from that of standard native peptide bonds in proteins, and the air/water interface also restricts the L -/ D -GAN molecular conformation. Therefore, Ramachandran angles is not suitable for the characterization of the structure of L -/ D -GAN assembly in our study. And, we will use β -sheet-like to describe the L -/ D -GAN supramolecular assembly structure at the interface. The detailed discussion was presented below.

[REDACTED]

The reason we prefer to use the β -sheet-like structure to describe the L-/D-GAn supramolecular assembly structure formed by the intermolecular hydrogen bonds of three amino acids in order to better elucidate the assembly pattern. The reasons are presented as follows.

1. **At present, the expression of β -sheet-like or helix-like is frequently used in the self-assembly studies of peptides.** Self-assembly of short peptides containing only 3-6 amino acid units to form stable secondary structures has been widely reported (*J. Am. Chem. Soc.* **2006**, *128* (41), 13539-13544; *Chem. Eur. J.* **2011**, *17* (46), 13095-13102; *Chem. Soc. Rev.* **2012**, *41*, 687-702.). For example, Lu group exhibited that ultrashort peptides I₃K (Ac-IIIK-NH₂) and L₃K (Ac-LLLK-NH₂) could readily self-assemble into nanotubes and nanosphere, respectively. The difference could arise from the different β -sheet promoting power between isoleucine and leucine, suggesting that while hydrophobic interaction was dominant in the formation of L₃K nanospheres hydrogen bonding governed the templating of antiparallel β -sheets and the subsequent formation of I₃K nanotubes (*Chem. Mater.* **2010**, *22*, 5165-5173.). Prof. Patrick Perlmutter et al. synthesized a series of N-Acetyl-Capped β -Peptides containing three or six amino acids. These peptides self-assemble in solution to form a 14-helix structure in an end-to-end arrangement through hydrogen bond interactions, and further assembled into nano- to macroscale fiber structure (*Angew. Chem. Int. Ed.* **2013**, *52*, 8266-8270.). Furthermore, the following studies also extend the concept of stable secondary structures to cyclic side-chain topology structures: (i) Prof. M. Reza Ghadiri et al. reported that under favour hydrogen bonding conditions, such as adsorption onto lipid membranes the Cyclic D, L- α -peptides can stack to form hollow, β -sheet-like tubular supramolecular structures that are open-ended (*Nature.* **2001**, *412*, 452-455.); (ii) Prof. Toshiyuki Moriuchi et al. reported that the introduction of only one minimum-sized peptide chain of the heterochiral sequence (-L-Ala-D-Pro-NH-4-Py

or -D-Ala-L-Pro-NH-4-Py) into the ferrocene scaffold can induce a type II β -turn-like structure based on the intramolecular hydrogen bonding (*Dalton Transactions*, **2009**, 95 (22), 4286-4288.); (iii) Prof. Bing Xu et al. reported a series of hydrogelators, which is the conjugation of an aromatic moiety to pentapeptides (GAGAS, GVPVP, VPGVG, VTEEL, VYGGG, and YGFGG). The self-assembly of the hydrogelators affords helical or β -sheet-like structures (*J. Am. Chem. Soc.*, **2010**, 132 (8), 2719-2728.); (iv) Prof. Santanu Bhattacharya et al. reported the short tripeptide sequence (Lys-Phe-Gly) self-assembles in water and the secondary structure of the tripeptide changes from random-coil to β -sheet-like assemblies as the concentration of tripeptide is increased (*Angew. Chem. Int. Ed.* **2014**, 53 (4), 1113-1117.); (v) Prof. Bing Xu et al. also showed that two complementary pentapeptides (RMLRF and IQEVN) from a β -sheet motif of a protein, being connected to an aromatic motif (i.e., pyrene) at their C-terminal, self-assemble to form β -sheet like structures upon mixing (*J. Am. Chem. Soc.* **2017**, 139 (1), 71-74.). (vi) Yang group synthesized two phosphorylated peptides, which could be converted to possible hydrogelators by enzyme-instructed self-assembly. At a physiological temperature of 37°C, the peptides would undergo fast transitions from random coil to -sheet- or -helix-like conformations, resulting in solution-to-gel transformations (*Chem. Commun.* **2019**, 55 (35), 5123-5126.).

The L-/D-GAn assembly formed regular structure with repeating segments and the hydrogen bonding pattern exhibits the properties like the β -sheet structure.

As shown in Figure. B1, in the β conformation, the backbone of the polypeptide chain is extended into a zigzag. The arrangement of several segments side by side, all of which are in the β conformation called a β -sheet. The zigzag structure of the individual polypeptide segments gives rise to a pleated appearance of the overall sheet. Hydrogen bonds form between adjacent segments of the polypeptide chain within the sheet. The amino-terminal to carboxyl-terminal orientations of adjacent chains (arrows) can be the opposite or the same, forming an antiparallel β -sheet or a parallel β -sheet. Hydrogen bonds between antiparallel β -sheets are essentially in-line. The ability of the three amide groups of the L-/D-GAn molecule to form inter-hydrogen bonds is the key to the

self-assembly of L/D -GAN into complex β -sheet-like structures. For clearly, we plotted schematic diagram (Figure. B2) of the hydrogen bond interactions between adjacent L/D -GAN molecules based on the SFG spectral results (Fig. 1e), combined with MD simulations (Fig. 3d) and AFM measurements (Supplementary Fig. 14). As shown in Figure. B2, an intermolecular hydrogen bond is formed between amide 3 group and the amide 2 group of the adjacent L/D -GAN molecules. L/D -GAN molecules are arranged in columns via intermolecular hydrogen bond and π - π stacking, and each column is like a β -sheet. Due to the hydrogen bonds formed by different amide groups, there is dislocation between the molecular columns, and the hydrogen bonds between the two columns of L/D -GAN molecules are essentially in-line. Therefore, the pattern of hydrogen bonds of L/D -GAN molecules assembled at the interface is very similar to that of antiparallel β -sheet structures.

Figure. B2 Schematic diagram of hydrogen bond interactions between amide 2 and amide 3 groups of adjacent L/D -GAN molecules. For the L/D -GAN assembly with an antiparallel β -sheet-like structure, the repeat period is 10 Å and the hydrogen bond (blue dashed lines) between adjacent L/D -GAN molecules are essentially in-line.

2. **The similar spectra features of antiparallel β -sheet-like is observed in our SFG experiment.** The chiral SFG spectra (Figs.1e, g, and Supplementary Fig. 6c) showed that the peaks centered at about $\sim 1630\text{ cm}^{-1}$ should be assigned to amide 2-3 (1637 cm^{-1}) and amide 3-3 (1628 cm^{-1}). Generally, the peak at 1637 cm^{-1} belongs to the B_2 mode of antiparallel β -sheet structure (*Chemical Communications*. **2016**,

52, 2956-2959; *Nanoscale*. **2012**, 4 (21), 6752-6760; *Q. Rev. Biophys.* **2002**, 35, 369-430; *Q. Rev. Biophys.* **1997**, 30, 365-429.). In addition, under the chiral polarization combination of psp and spp, the amide I of amide 3 showed the same intensity, that is

$$\chi_{eff,psp}^{(2)} = \chi_{eff,spp}^{(2)}$$

(Supplementary Table 5), which is consistent with the derived equation

$$\chi_{eff,psp}^{(2)} = \chi_{eff,spp}^{(2)} = -\frac{1}{2}N_s(\cos^2\theta - \sin^2\theta\cos^2\psi)\beta_{acb}$$

for B₂ mode of antiparallel β -sheet structures in the Supplementary Section 1.4.

In conclusion, all the discussions above confirmed that the L-/D-GAn formed a significant complex antiparallel β -sheet-like supramolecular chirality structure at the air/water interface.

Therefore, we added the following highlighted sentences in Supplementary Section 5, and replaced Supplementary Fig. 7 with Figure. B2.

In this study, L-/D-GAn molecules are not self-assembled into standard secondary structures as proteins or polypeptides because the L-/D-GAn self-assembly at the interface do not have standard peptide chains and are restricted by the interface. However, the L-/D-GAn supramolecular assembly structure still exhibits somewhat antiparallel β -sheet-like structure, because (i) the similar chiral SFG peak position of antiparallel β -sheet-like (amide I band of amide 2-3, 1637 cm⁻¹) is observed, as shown in Supplementary Fig. 6; (ii) the L-/D-GAn assembly shows a regular structure with repeating segments, and the hydrogen bonding pattern exhibits similar properties to that of the β -sheet structure, as shown in Supplementary Fig. 7. It is known that, in the β -sheet conformation, the backbone of the polypeptide chain is extended into a zigzag. The arrangement of several segments side by side, all of which are in the β conformation, is called a β -sheet. Hydrogen bonds form between adjacent segments of the polypeptide chain within the sheet and are essentially in-line in an antiparallel β -sheet structure. For L-/D-GAn assembly, L-/D-GAn molecules are arranged in columns via intermolecular

hydrogen bond and π - π stacking. Each column is formed like a β -sheet, and the hydrogen bonds between the two columns of L-/D-GAn molecules are essentially in-line. The pattern of hydrogen bonds of L-/D-GAn molecules assembled at the interface is very similar to that of antiparallel β -sheet structures in protein or polypeptides. Therefore, we conclude that L-/D-GAn molecules self-assemble at the air/water interface to form antiparallel β -sheet-like supramolecular structures.

The main text has been changed as below:

The original sentence on page 6 in the main text has been changed from “Because L-/D-GAn molecules are not connected in the same way as standard native peptide bonds, for clarity, we redefine and calculate the Ramachandran angles in chiral supramolecular assembly systems by following the definition of Ramachandran angles in peptide and protein systems, as shown in Supplementary Fig. 7, which confirms the results of chiral SFG spectroscopy that the L-/D-GAn assembly forms an antiparallel β -sheet-like structure.” to “Since L-/D-GAn molecules are linked differently from standard natural peptide bonds, for clarity, a schematic diagram of the hydrogen bonding links of the L-/D-GAn supramolecular assembly was drawn based on the results of chiral SFG spectroscopy and MD simulations (Supplementary Fig. 7), which exhibit an antiparallel β -sheet-like structure in the L-/D-GAn assembly.”

REVIEWERS' COMMENTS

Reviewer #2 (Remarks to the Author):

The point-to-point response and revisions look good to the reviewer, and I would recommend to accept the article.

Reviewer #3 (Remarks to the Author):

The authors have corrected the mistake and modified the discussion about protein secondary structures. The authors now use the term "beta-sheet-like" structure, which is acceptable. The manuscript can be accepted as is.